# mTORC1 and mTORC2 regulate skin morphogenesis and epidermal barrier formation

Xiaolei Ding[1,2], Wilhelm Bloch[3], Sandra Iden[2,4], Markus A. Rüegg[5], Michael N. Hall[5], Maria Leptin[2,6,7], Linda Partridge[4,8,9] & Sabine A. Eming[1,2,4]

Mammalian target of rapamycin (mTOR), a regulator of growth in many tissues, mediates its activity through two multiprotein complexes, mTORC1 or mTORC2. The role of mTOR signalling in skin morphogenesis and epidermal development is unknown. Here we identify mTOR as an essential regulator in skin morphogenesis by epidermis-specific deletion of *Mtor* in mice (mTOR[EKO]). mTOR[EKO] mutants are viable, but die shortly after birth due to deficits primarily during the early epidermal differentiation programme and lack of a protective barrier development. Epidermis-specific loss of *Raptor*, which encodes an essential component of mTORC1, confers the same skin phenotype as seen in mTOR[EKO] mutants. In contrast, newborns with an epidermal deficiency of *Rictor*, an essential component of mTORC2, survive despite a hypoplastic epidermis and disruption in late stage terminal differentiation. These findings highlight a fundamental role for mTOR in epidermal morphogenesis that is regulated by distinct functions for mTORC1 and mTORC2.

[1] Department of Dermatology, University of Cologne, Kerpenerstr. 62, Cologne 50937, Germany. [2] Center for Molecular Medicine Cologne (CMMC), University of Cologne, Robert-Koch-Str. 21, Cologne 50931, Germany. [3] Department of Molecular and Cellular Sport Medicine, German Sport University Cologne, Am Sportpark Müngersdorf, Cologne 50933, Germany. [4] Cologne Excellence Cluster on Cellular Stress Responses in Aging-Associated Diseases (CECAD), University of Cologne, Joseph-Stelzmann-Str. 26, Cologne 50931, Germany. [5] Biozentrum, University of Basel, Klingelbergstrasse 50/70, Basel CH-4056, Switzerland. [6] Institute for Genetics, University of Cologne, Zülpicherstr. 47a, Cologne 50674, Germany. [7] European Molecular Biology Laboratory, Meyerhofstr. 1, Heidelberg 69117, Germany. [8] Max Planck Institute for Biology of Ageing, Joseph-Stelzmann-Str. 9b, Cologne 50931, Germany. [9] Institute of Healthy Ageing, Department of Genetics, Evolution, and Environment, University College London, London WC1E 6BT, UK. Correspondence and requests for materials should be addressed to S.A.E. (email: sabine.eming@uni-koeln.de).

The epidermis as the outer layer of the skin serves as a primary interface between the body and its environment and protects the organism from dehydration and external insults. The epidermis is a stratified squamous epithelium that fulfils its functions through a lifelong self-renewal process that is precisely coordinated by regenerative pathways, which in part recapitulate those that are activated in epidermal morphogenesis[1]. Therefore, elucidating mechanisms of skin development provides the understanding for molecular principles of skin physiology and disease development throughout life.

A complex network of signalling pathways and factors contribute to skin morphogenesis, epidermal stratification and protective barrier formation including p63, Wnt, Notch and the PI3K-Akt cascade[2,3]. These factors govern the dynamics between progenitor cell division in the epidermal basal layer, delamination, oriented cell divisions and terminal differentiation of daughter cells within suprabasal layers[4]. The exact mechanisms that orchestrate this fine-tuned balance between cell proliferation and commitment to differentiation remain to be determined[5–7]. The role of mammalian target of rapamycin (mTOR), a regulator of cell growth and proliferation, has not been investigated in epidermal morphogenesis. mTOR is a conserved serine-threonine kinase that acts primarily via the regulation of protein synthesis[8,9]. Multiple upstream signals regulate the mTOR pathway, including the 'classical' mTOR regulators, such as the receptor tyrosine kinase-PI3K-Akt signalling cascade[10,11], and the more recently described inputs from keratin 17 or the Wnt and Notch pathways[12–15]. These pathways are critical in epidermal morphogenesis, and we therefore hypothesized that mTOR might serve a role in skin development and epidermal barrier formation.

mTOR mediates its functions through the assembly of two structurally distinct multiprotein complexes, mTORC1 and mTORC2 (refs 10,11,14). The regulatory-associated protein of mTOR (Raptor) and the Rapamycin-insensitive companion of mTOR (Rictor) are both mTOR-associated adaptor proteins, and are required for the formation and function of mTORC1 and mTORC2, respectively. mTORC1 regulates mRNA translation by phosphorylating the translational regulators ribosomal protein S6 kinases (S6K 1 and 2) and eukaryotic initiation factor 4E-binding protein 1 (4E-BP1)[16–18]. Functions of mTORC2 are less well defined than those of mTORC1, and one of the best characterized downstream targets is Akt/protein kinase B, which is phosphorylated on S473, and which contributes primarily to maintenance of cellular size and viability[19–21]. In addition, there is evidence for a role for mTORC2 in remodelling of actin architecture through PKCα and Rac GTPase[22,23].

Germline deletion of Mtor or genes encoding essential components of mTOR complexes, such as Rptor or Rictor leads to early embryonic lethality in various organisms[21,24]. The recent availability of conditional mouse lines has made it possible to address cell-type-specific functions of mTOR signalling components and has revealed their roles in homeostasis of a wide range of organs[25,26]. Knowledge on the function and regulation of mTOR pathway components in skin maintenance and homeostasis is limited. In pre-clinical and clinical studies, roles of mTOR signalling in wound healing of cutaneous or mucosal injuries[15,27,28], epidermal stem cell homeostasis[13] and epidermal carcinogenesis[29,30], have been described. Whereas mTOR inhibitors appear to be tolerated by healthy skin, wound complications are one of the most frequent side-effects of mTOR-inhibitors used in the clinic and can lead to therapy interruption[31].

Here we examine the role of the mTOR pathway in skin morphogenesis in mice by inactivating mTOR or its adaptor proteins Raptor or Rictor specifically in the epidermis. The findings revealed essential and distinct functions of mTORC1 and mTORC2 in skin morphogenesis that have implications for the understanding of skin physiology.

## Results

**mTOR[EKO] mutants fail to form a protective epidermis.** To determine the role of mTOR signalling in the epidermal compartment of the skin, we specifically deleted Mtor in the epidermis. Mice carrying a loxP-flanked allele encoding Mtor (Mtor[fl/fl])[25] were crossed with a transgenic mouse line expressing Cre recombinase under the control of the human keratin 14 (K14) promoter[32], leading to epidermis-specific Mtor deletion (mTOR[EKO]) in the progeny (Supplementary Fig. 1a). Cre-mediated recombination of Mtor floxed alleles was verified by PCR analysis of genomic DNA extracted from newborn mouse epidermis, and resulted in effective reduction of mTOR protein expression in embryonic epidermis as revealed by western blotting analysis (E17.5) and immunohistochemical staining (E15.5) (Fig. 1a,b). mTOR[EKO] mice were born (P0) with a highly fragile and translucent skin morphology (Fig. 1c). After birth, pups showed signs of severe dehydration and died rapidly within a few hours (Fig. 1d; Supplementary Fig. 1b). Except for the obvious skin abnormalities, mTOR[EKO] newborns were comparable to their littermate controls in body size and length of limbs (Fig. 1c; Supplementary Fig. 1b). The expected Mendelian ratio was observed with newborns and embryos during gestation (Supplementary Fig. 1c). Pups that were heterozygous for the Mtor deficiency in the epidermis (Mtor[fl/wt]K14Cre mice) appeared normal and survived until adulthood without any obvious phenotype (Supplementary Fig. 1d,e).

Consistent with the severe macroscopic alterations, histology of newborn skin in mTOR[EKO] mice revealed severe abnormalities (Fig. 2a). The epidermis in controls showed a stratified epithelium consisting of basal, spinous, granular and cornified layers, whereas the epidermis in mTOR[EKO] mice was reduced to a 1–2 cell layer epithelium lacking signs of stratification (Fig. 2a). The thickness of the dermis, and both the number and development of hair follicles were markedly reduced in mTOR[EKO] pups (Fig. 2a,b). Also the epithelium of the tongue

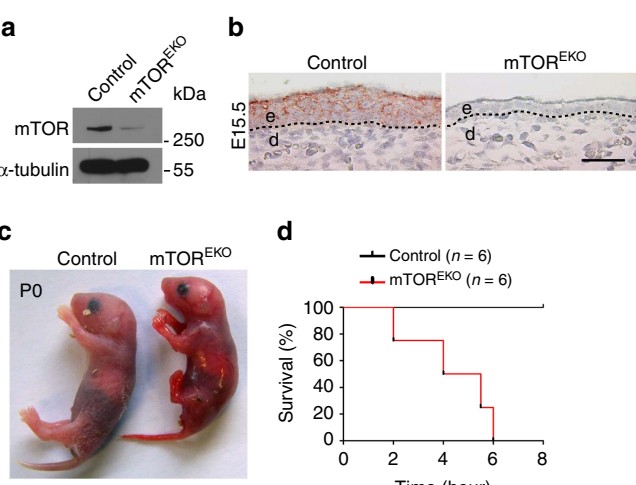

**Figure 1 | Conditional targeting of Mtor and perinatal lethality of mTOR[EKO] mice.** (a) Western blot analysis for mTOR protein in epidermis isolated from control and mTOR[EKO] embryos. (b) Representative mTOR-immunostaining of skin in embryos at E15.5; dashed line indicates basal membrane; scale bar 25 μm; (c) Macroscopic appearance of newborns. (d) Kaplan–Meier plot illustrating the survival rate of newborns.

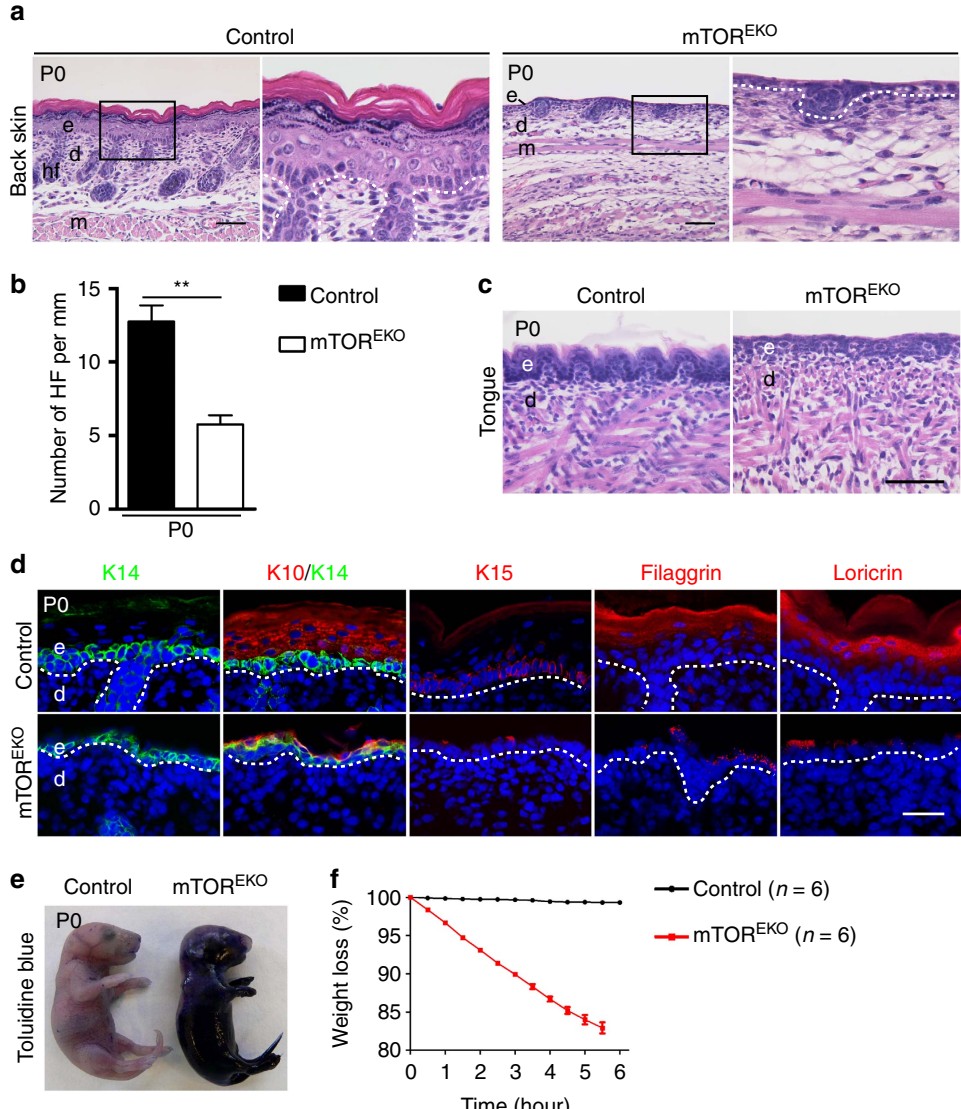

**Figure 2 | mTOR^EKO mutants are viable but fail to form a stratified and protective epidermis.** (**a**) Representative H&E-stained back skin in newborns. (**b**) Quantification of hair follicles (HF) on back skin (n = 5 mice/genotype). (**c**) Representative H&E-stained tongue of newborns. (**d**) Representative immunostaining of epidermal markers: keratin 14 (K14, green), keratin 10 (K10, red), keratin 15 (K15, red), filaggrin (red) and loricrin (red) of back skin in newborns, (DAPI stain, blue), dashed line indicates basal membrane. (**e**) Toluidine blue dye penetration assay with newborn mice. (**f**) Perinatal weight loss of newborns. e, epidermis; d, dermis; hf, hair follicle; m, muscle; scale bar, (**a**,**c**) 50 µm, (**d**) 25 µm. Data represents mean ± s.d.; non-paired t-test was used to calculate P value. **P < 0.01.

in mTOR^EKO newborns lacked stratification when compared with controls (Fig. 2c).

In control mice the cells of the basal layer stained positive for keratin 14 (K14), and suprabasal cells were positive for stratification and differentiation markers including keratin 10 (K10), filaggrin and loricrin (Fig. 2d). In contrast, in mTOR^EKO mice the basal cell layer stained positive for K14, and individual cells above it stained occasionally positive for K10, but markers for terminal differentiation including filaggrin and loricrin, were virtually absent (Fig. 2d). Keratin 15 (K15) as a marker for epidermal stem cells and progenitors was detected in controls but was absent in mutant mice (Fig. 2d).

The alterations in epidermal stratification in mTOR^EKO newborns were associated with severe disturbances in barrier function. Whereas in newborn controls exogenously applied Toluidine blue dye did not penetrate the skin, in mTOR^EKO newborns the dye easily penetrated the skin and the mice appeared blue (Fig. 2e). Consistent with a loss of barrier function

in mTOR^EKO mice, mutants lost up to 20% of their initial body weight within 6 h after birth (Fig. 2f). Together, these results show severely disturbed functions of both the 'outside-in' and the 'inside-out' skin barrier function in mTOR^EKO mice. Furthermore, epidermal-mesenchymal communication, which is required for morphogenesis of hair follicles appeared severely disturbed.

**mTOR^EKO embryos fail to form a stratified epidermis.** To further dissect the role of mTOR in epidermal morphogenesis and barrier development, we tested barrier function and performed histological analyses of skin from embryos at various developmental stages starting from E15.5. Epidermal barrier function is normally acquired in this late embryonic period, coinciding with the programme of stratification and hair follicle formation[2–6,33]. Macroscopically, E15.5 mTOR^EKO embryos were indistinguishable from their littermate controls, but by E16.5 and

E17.5 the skin of mTOR[EKO] embryos was fragile and appeared shiny (Fig. 3a). As revealed by the Toluidine blue dye penetration test at E16.5, the epidermis of control embryos initiated a barrier formation programme at the dorsum, preventing outside-in penetration of the dye, which at E17.5 was almost complete (Fig. 3b). In contrast, at all stages the epidermis in mTOR[EKO] embryos failed to form a functional barrier and the dye penetrated the entire embryonic skin (Fig. 3b).

Histological analysis of controls showed the development of a stratified epithelium with formation of a stratum corneum by E17.5 (Fig. 3c). In contrast, E15.5 mTOR[EKO] embryonic skin consisted of a thin layer of flattened basal epithelial cells and a

further suprabasal layer losing the typical cuboidal shape of basal epithelial cells in E15.5 control skin (Fig. 3c). E16.5 mTOR[EKO] epidermis appeared hypoplastic and disorganized and had not stratified further by E17.5. Consistently, qPCR analysis of epidermal tissue at E17.5 revealed attenuated expression of both basal and suprabasal markers in mTOR[EKO] embryos (Supplementary Fig. 2a). To further characterize the developmental abortion and the fate of cells in which the stratification programme arrests, we performed in epidermis of E17.5 mTOR[EKO] mutants mRNA expression and immuno-histochemical analyses of simple epithelia markers keratin 8 (K8) and keratin 18 (K18). Expression of K8/18 was significantly

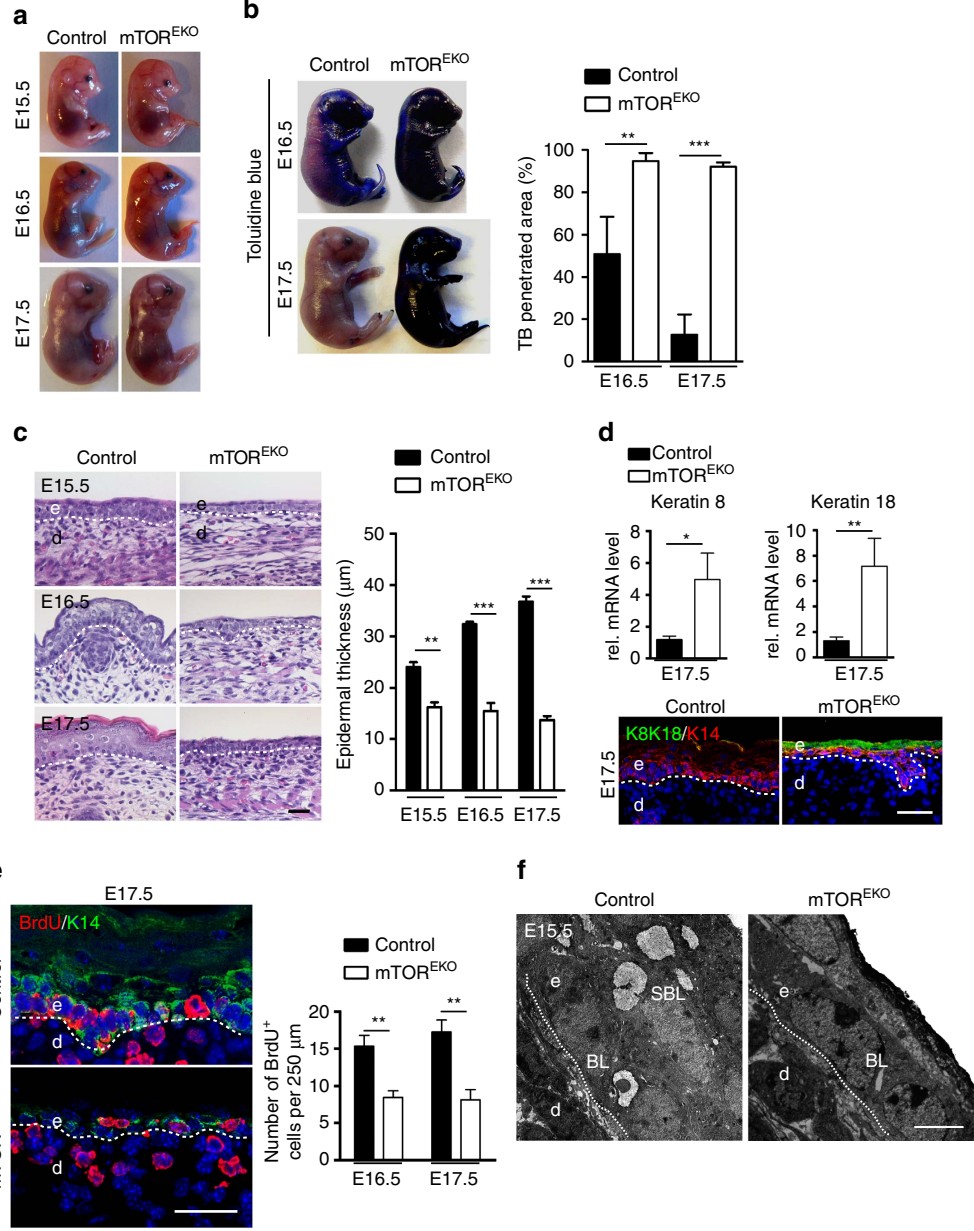

**Figure 3 | mTOR[EKO] embryos fail to initiate epidermal stratification.** (**a**) Macroscopic appearances of mTOR[EKO] and control embryos. (**b**) Toluidine blue dye penetration assay and the quantification of the blue stained area ($n = 5$ embryos/genotype). (**c**) Representative H&E-stained back skin of embryos and the quantification of epidermal thickness at indicated time points ($n = 5$ embryos/genotype). (**d**) qRT–PCR analysis and representative immunostaining of keratin 8/18 (K8/K18) in mTOR[EKO] and control epidermis at E17.5 ($n = 5$ embryos/genotype). (**e**) BrdU-immunostaing and quantification of basal BrdU[+] cells in embryo epidermis at E16.5 and E17.5 ($n = 5$ embryos/genotype). (**f**) Representative TEM of epidermis at E15.5 reveals an atypical epithelial cell morphology of basal cells with loss of the typical cubical morphology recognizable in E15.5 control epithelium ($n = 3$ embryos/genotype). Scale bar, (**c–e**) 25 μm, (**f**) 2 μm. Data represents mean ± s.d.; non-paired $t$-test was used to calculate $P$ value. *$P < 0.05$, **$P < 0.01$, ***$P < 0.001$.

increased in mutants versus controls, and co-immunostaining with K14 revealed that K8/18 was expressed in basal and suprabasal cell layers, the latter most likely representing the residual periderm (Fig. 3d). The fact that in mutants K8/18-K14-double-positive cells are maintained suggests that in mTOR-deficient cells the keratin switch is delayed/impaired, and halted at a stage where both keratins are expressed.

In control embryos, hair follicle formation started to be visible at E16.5 (Fig. 3c) and hair buds stained positive for P-cadherin at this stage (Supplementary Fig. 2b)[34]. As judged by P-cadherin expression in E17.5 epidermis, the number of hair follicles in mTOR[EKO] embryos was significantly reduced and their maturation delayed suggestive for disturbed epidermal–mesenchymal interactions (Supplementary Fig. 2b).

The morphological and functional abnormalities of mTOR[EKO] embryonic skin resembled those described in p63-null mice[35,36]. p63 is a transcription factor that is expressed in basal keratinocytes, that controls progenitor cell proliferation and epidermal stratification[4,37,38]. Immunostaining and western blotting analyses showed attenuated p63 expression in mTOR[EKO] epidermis (Supplementary Fig. 2c). qRT–PCR analysis further confirmed that the mRNA expression of ΔNp63 isoforms was significantly reduced in mTOR[EKO] epidermis versus controls. Consistently, mTOR[EKO] epidermis also showed reduced expression of Irf6, Gata3 and Ikkα, and upregulated expression of Runx2, which represent activated or repressed targets by p63, respectively, that are critical in epidermal morphogenesis (Supplementary Fig. 2c)[39–42].

To examine whether the hypoplastic epidermis in mTOR[EKO] mice resulted from reduction in cell proliferation, we performed BrdU incorporation assays with E16.5 and E17.5 embryos. Both in control and mutant mice incorporation of BrdU was detectable

in the cells of the basal layer (Fig. 3e). However, the number of BrdU[+] cells was reduced by 42.2% ± 12.7 and 52% ± 17.2 in mTOR[EKO] embryos at E16.5 and E17.5, respectively. To clarify whether apoptosis contributed to epidermal hypoplasia in mTOR[EKO] embryos, we performed TUNEL assays, caspase-3 stainings and transmission electron microscopy (TEM) analysis. No obvious signs of apoptosis in the skin of mTOR[EKO] embryos or controls were found (Fig. 3f; Supplementary Fig. 2d). In summary, the defects observed in the skin of newborn mTOR[EKO] mice developed during the early epidermal differentiation programme, were in part caused by defects in proliferation and may involve disturbed p63 signalling.

**mTOR deficiency alters downstream mTORC1 and mTORC2 signalling.** To determine whether mTOR function in epidermal morphogenesis is mediated by mTORC1 or mTORC2 we examined downstream targets for each multiprotein complex. The epidermis of E17.5 mTOR[EKO] and control embryonic skin was separated and analysed by western blotting analysis (Fig. 4a). Expression levels of Raptor and Rictor were comparable in controls and mutants. In contrast, the phosphorylation of the mTORC1 targets S6K at T389, S6 at S240/244 and 4E-BP1 at T37/46 was markedly reduced in the epidermis of mTOR[EKO] embryos (Fig. 4a). The phosphorylation of the mTORC2 target Akt at S473 was also substantially attenuated. The phosphorylation of FoxO1 at T24 and S256, of GSK3α at S21 and GSK3β at S9, all known downstream targets of Akt-pS473, were comparable in controls and mutants (Fig. 4a). These findings demonstrate that genetic loss of *Mtor* in the epidermis attenuates signalling pathways of both mTORC1 and mTORC2.

We also performed immunohistochemical stainings in embryonic skin at E16.5 or E17.5 for S6-pS240/244, 4E-BP1-pT37/46 and

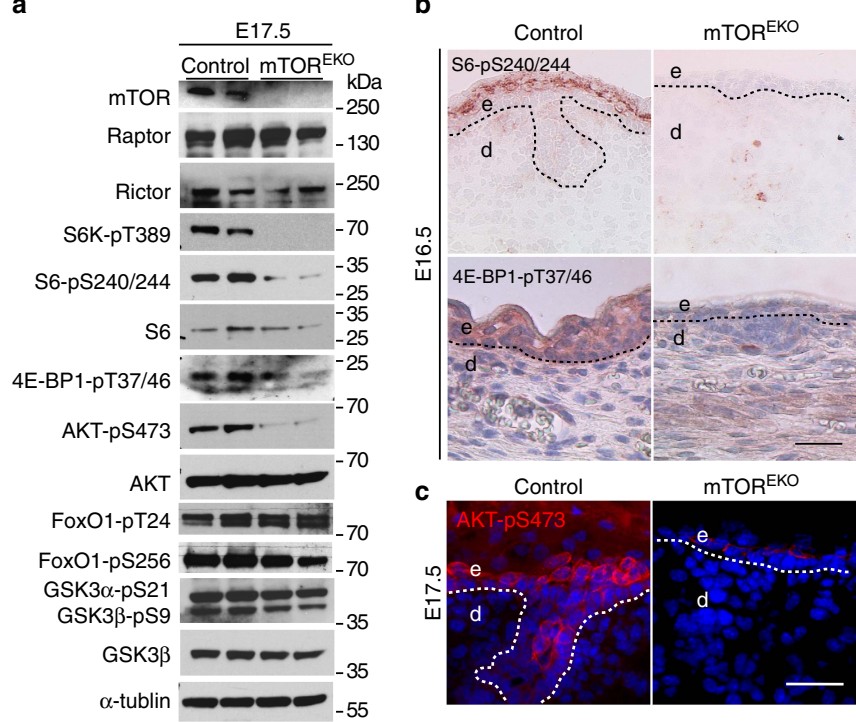

**Figure 4 | mTORC1 and mTORC2 are activated during epidermal embryogenesis but downstream targets are attenuated in mTOR[EKO] mice.**
(**a**) Representative western blotting analysis from three independent epidermal extract preparations at E17.5 with antibodies against indicated proteins.
(**b**) Immunostaining for pS6 (S6-pS240/244) and p4E-BP1 (4E-BP1-pT37/46) of back skin in embryos at E16.5. (**c**) Phospho-Akt (Akt-pS473) immunostaining on back skin in embryos at E17.5; dashed line indicates basal membrane; scale bar, (**b,c**) 25 μm.

Akt-pS473 (Fig. 4b,c). In control epidermis a clear signal for S6-pS240/244 and 4E-BP1-pT37/46 was detected in the suprabasal or basal epidermal layers starting at E16.5, respectively, suggesting activation of mTORC1 during epidermal stratification. pS6 and p4E-BP1 staining was virtually undetectable in mutant embryonic epidermis (Fig. 4b). Consistent with previous observations, phosphorylation of Akt at S473 became detectable in control spinous and granular layers at E17.5 (Fig. 4c)[43]. In mTOR[EKO] embryonic epidermis phosphorylation of Akt-S473 was significantly reduced, although in individual cells above the stratum basale staining was occasionally detected (Fig. 4c).

**mTOR controls proliferation and differentiation through mTORC1.** To dissect the roles of the two mTOR complexes in epidermal morphogenesis, we generated mice with epidermis-specific deletion of *Rptor* or *Rictor*[26] (Fig. 5a; Supplementary Fig. 3a). Newborn Rap[EKO] mice were viable but the skin was fragile, shiny, translucent and the barrier function was severely compromised (Fig. 5b,c). Like mTOR[EKO] pups, newborn Rap[EKO] mice showed signs of severe dehydration and died rapidly within a few hours of birth (Supplementary Fig. 3b).

Histological analysis of newborn Rap[EKO] mouse skin revealed multiple defects that resembled those seen in mTOR[EKO] mice: absence of the normal epidermal stratification in both back skin and tongue (Fig. 5d; Supplementary Fig. 3c); a hypoplastic dermis and a significantly reduced number and development of hair follicles in the back skin (Fig. 5d); absence of terminal differentiation markers such as filaggrin and loricrin, whereas

K14 was detectable, as was K10, albeit at minimal levels; absence of K15 staining (Fig. 5e).

Furthermore, as in mTOR[EKO] embryos, epidermal morphogenesis in Rap[EKO] embryos was severely impaired as shown by a translucent skin (Fig. 6a) and increased penetration of Toluidine blue dye at E17.5 (Fig. 6b), loss of epidermal stratification at E16.5 and E17.5 (Fig. 6c), attenuated formation of hair follicles and retained expression of K8/18 in K14-positive basal cells and in periderm (Supplementary Fig. 3d,e). Similarly, expression of ΔNp63 isoforms and their target genes Irf6 and Gata3 were downregulated, and Runx2 upregulated (Supplementary Fig. 3f). The BrdU incorporation assay revealed significantly fewer proliferative cells in the basal layer of Rap[EKO] embryos (Fig. 6d).

mTORC1 has key roles in mRNA translation[10,11]. To examine whether the disturbed epidermal differentiation phenotype in Rap[EKO] embryos might in part be caused on the translational level, we compared differentiation marker RNA and protein expression levels in epidermal extracts of Rap[EKO] and control embryos (E17.5). qRT–PCR and western blotting analysis of epidermal tissues showed a significant reduction of differentiation markers (K1, K10, loricrin, filaggrin) at the transcriptional and also at the protein level in Rap[EKO] mutants when compared with controls (Fig. 6e,f). Hence, based on these findings we have no evidence that aborted epidermal differentiation in TORC1-deficient epidermal tissues is particularly due to reduced translation of the tested differentiation markers.

To determine which downstream targets of mTORC1 could contribute to epidermal morphogenesis we performed western blotting analysis in E17.5 embryonic skin of Rap[EKO] mice. Whereas

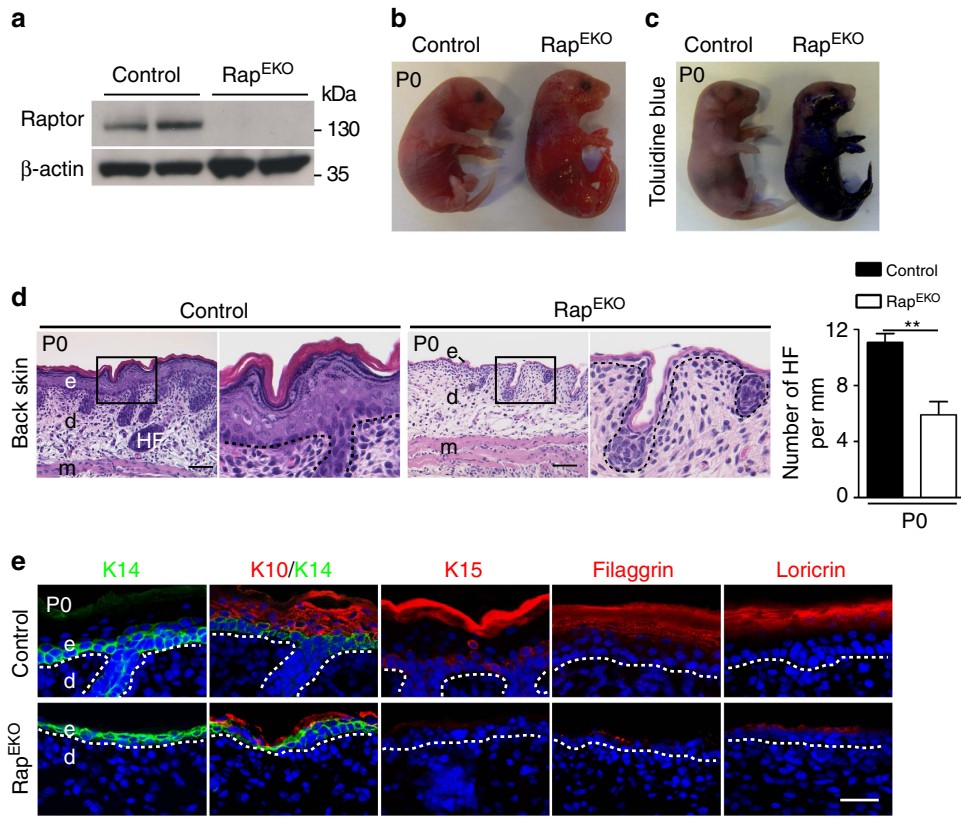

**Figure 5 | Epidermal-specific deletion of *Rptor* phenocopies mTOR[EKO] mice.** (**a**) Representative western blotting analysis for Raptor protein in epidermis isolated from control and Rap[EKO] newborns. (**b**) Macroscopic appearance and (**c**) Toluidine blue dye penetration assays with newborn mice. (**d**) Representative H&E-stained back skin in newborns; quantification of hair follicles (HF) on back skin (*n* = 5 mice/genotype). (**e**) Representative immunostaining of epidermal markers: keratin 14 (K14, green), keratin 10 (K10, red), keratin 15 (K15, red), filaggrin (red) and loricrin (red) of back skin in newborns, (DAPI stain, blue), dashed line indicates basal membrane; scale bar, (**d**) 50 μm, (**e**) 25 μm. Data represents mean ± s.d.; non-paired *t*-test was used to calculate *P* value. **P < 0.01.

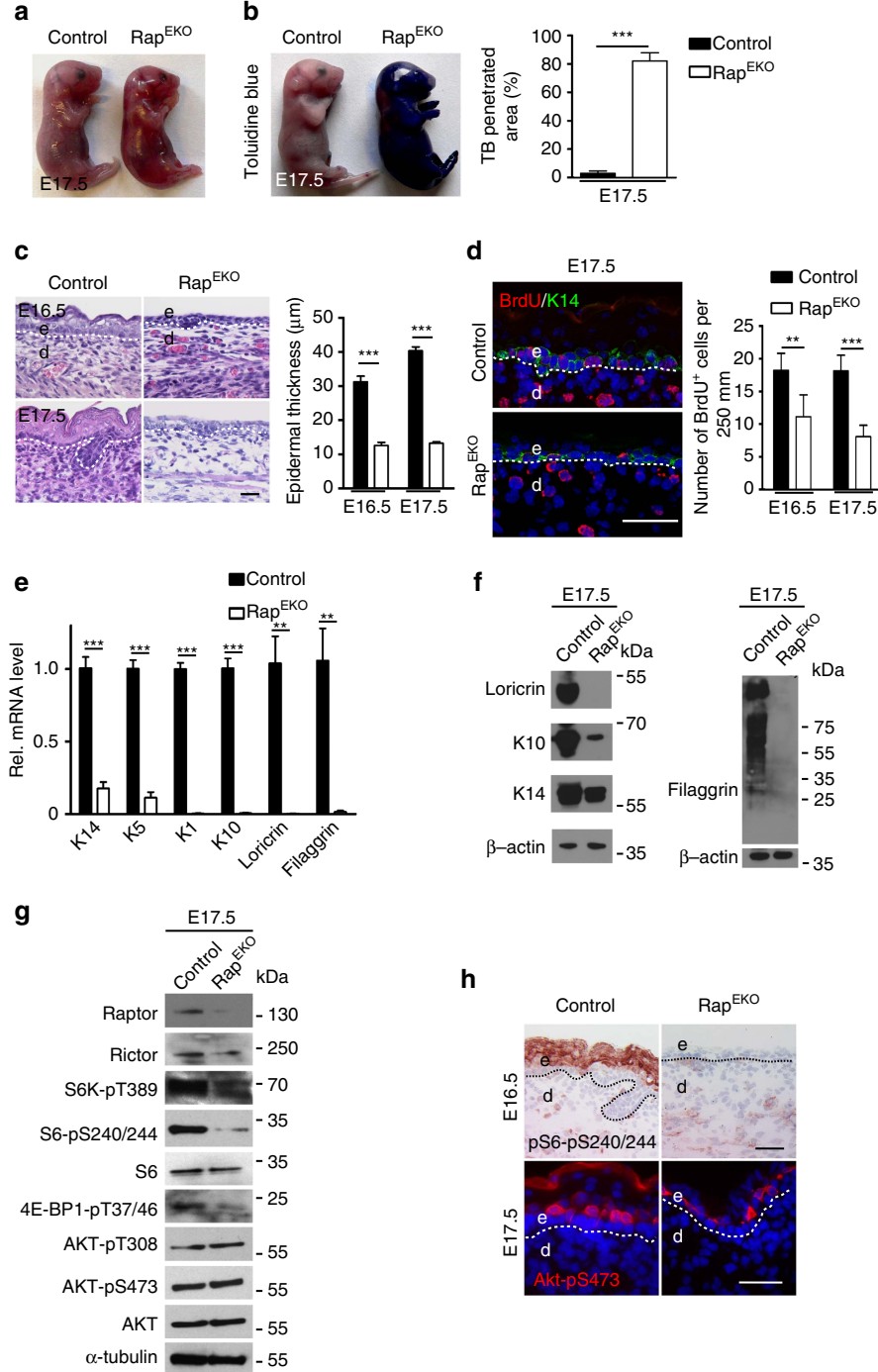

**Figure 6 | Rap^EKO mutants are viable but fail to initiate epidermal stratification.** (**a**) Macroscopic appearances and (**b**) Toluidine blue dye penetration assay of embryos at E17.5, and the quantification of the blue stained area ($n = 5$ embryos/genotype). (**c**) Representative H&E-stained back skin of embryos and the quantification of epidermal thickness at E16.5 and E17.5. (**d**) BrdU$^+$ (red) and K14 (green) double-immunostaing and quantification of basal BrdU + cells in E16.5 and E17.5 epidermis. (**e**) qRT–PCR analysis and (**f**) western blotting analysis in Rap^EKO and control epidermis at E17.5 ($n = 5$ embryos/genotype). (**g**) Representative western blotting analysis of three independent epidermal extract preparations at E17.5 with antibodies against indicated proteins. (**h**) Representative phospho-S6 (S6-pS240/244) and phospho-Akt (Akt-pS473) immunostaining of back skin in embryos at E16.5. Scale bar, (**c,d,h**,) 25 μm. Data represents mean ± s.d.; non-paired $t$-test was used to calculate $P$ value. **$P < 0.01$, ***$P < 0.001$.

Raptor protein was absent, expression of Rictor was not affected. Phosphorylation of the mTORC1 targets S6K, S6 and 4E-BP1 was reduced in mutants (Fig. 6g). In contrast, we did not detect obvious alterations of phosphorylated Akt on residues T308 and S473 in mutants (Fig. 6g). Disturbances of the mTORC1 pathway in Rap^EKO mice during epidermal morphogenesis were further corroborated by immunohistochemistry of E16.5 and E17.5 embryonic epidermis. Whereas phosphorylation of pS6-pS240/244 and Akt-pS473 was easily detectable in suprabasal cell layers of control embryos, in Rap^EKO embryos phosphorylation for S6-pS240/244 was absent but Akt-pS473 was detectable in the few suprabasal cells present in Rap^EKO epidermis (Fig. 6h).

**mTORC2 controls stratification and barrier formation**. We further generated epidermal-specific *Rictor* knockout mice (Ric^EKO) (Fig. 7a; Supplementary Fig. 4a). Ric^EKO mice were born at the expected Mendelian ratio and presented a body size that was comparable to control littermates (Fig. 7b; Supplementary Fig. 4b). Although the majority of Ric^EKO newborns survived through adulthood, up to 20% of the mice died within 3 months. The skin of newborn Ric^EKO mice was characterized by a shiny, translucent and more finely wrinkled morphology (Fig. 7b). Although, the macroscopic defects in Ric^EKO newborns were less severe than those in mTOR^EKO and Rap^EKO pups, they were nevertheless clearly distinguishable from control littermates (Fig. 7b; Supplementary Fig. 4b).

The epidermis of Ric^EKO newborn skin was stratified, but the epidermal thickness was reduced by 46% ± 4. This was primarily due to a reduced granular cell layer and thinner stratum corneum (Fig. 7c,d). Analysis of newborn skin by TEM did not reveal obvious alterations at the subcellular level (Supplementary Fig. 4c). Also the tongue displayed a hypoplastic epithelium in Ric^EKO newborns (Supplementary Fig. 4d). The dermal thickness and both the number and size of hair follicles were comparable between the control and Ric^EKO mice (Fig. 7c,d). Toluidine blue dye penetration assays showed that the outside-in barrier function was overall intact, although occasionally, Ric^EKO pups showed small erosions, which might have occurred due to mechanical stress of the skin during birth (Fig. 7e) perhaps indicating skin fragility in the mutant mice. Trans-epidermal water loss in Ric^EKO mice was increased 1.5-fold compared with controls (Fig. 7f), suggesting an impairment of the epidermal inside-out barrier function.

Immunohistochemical staining for K14 and K15 in the basal cell layer was comparable in mutants and controls (Fig. 7g). The differentiation markers K10, loricrin and filaggrin were detected in Ric^EKO pups although the overall abundance was reduced when compared with controls consistent with the epidermal hypoplasia in Ric^EKO mutants (Fig. 7g).

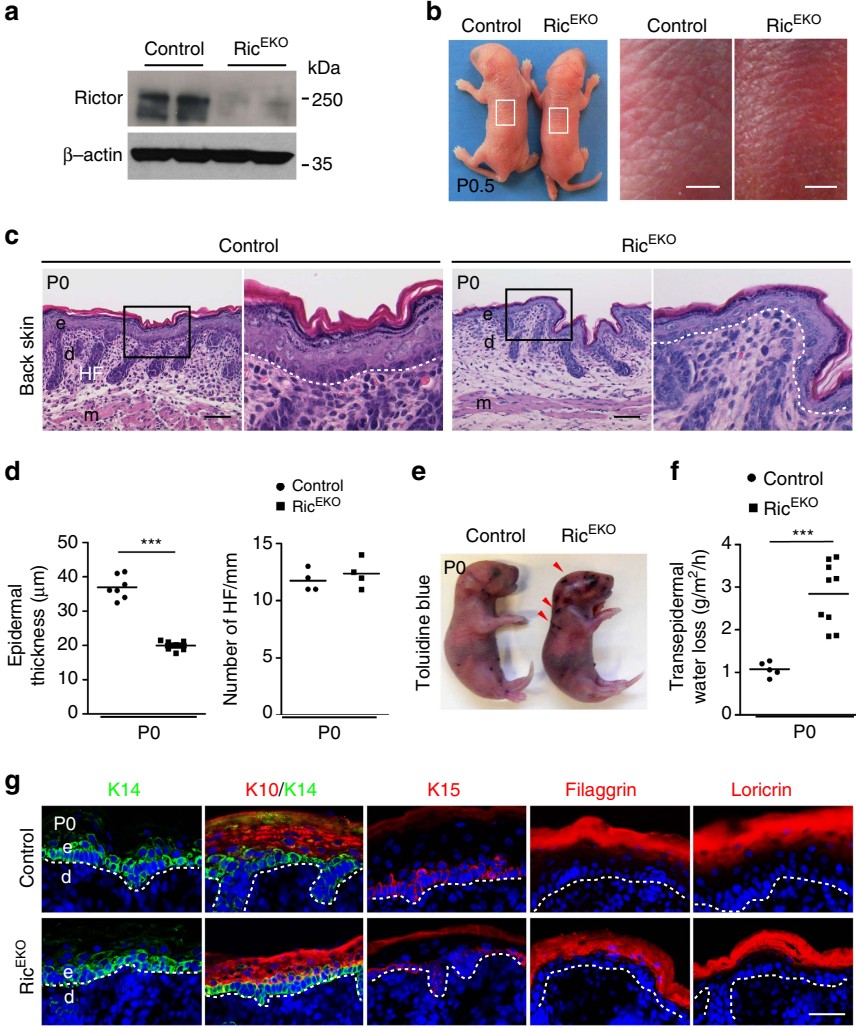

**Figure 7 | mTORC2 controls stratification and epidermal barrier formation of the interfollicular epidermis.** (**a**) Western blot analysis for Rictor protein in epidermis isolated from control and Ric^EKO embryos. (**b**) Representative macroscopic appearance of newborns; note fine-wrinkled texture in Ric^EKO mice. (**c**) Representative H&E-stained back skin in newborns. (**d**) Quantification of epidermal thickness and hair follicles (HF) on back skin (each dot represents one mouse). (**e**) Toluidine blue dye penetration assays with control and Ric^EKO newborn mice; arrowheads indicate skin lesions and penetration of dye. (**f**) Quantification of TEWL in newborns (each dot represents one mouse). (**g**) Representative immunostaining of epidermal markers: keratin 14 (K14, green), keratin 10 (K10, red), keratin 15 (K15, red), filaggrin (red) and loricrin (red) of back skin in newborns, (DAPI stain, blue), dashed line indicates basal membrane. e, epidermis; d, dermis; HF, hair follicle; m, muscle; scale bar: (**b**) 2 mm, (**c**) 50 μm, (**g**) 25 μm. Data represents mean ± s.d.; non-paired *t*-test was used to calculate *P* value. ***P < 0.001.

**mTORC2 regulates differentiation and asymmetric cell division**. To examine the role of mTORC2 in skin morphogenesis Ric$^{EKO}$ embryos were collected at E17.5 and E18.5 and subjected to the Toluidine blue dye penetration assay. Whereas blue staining in control embryos at E17.5 was limited to the distal limbs and defined areas of the head, in Ric$^{EKO}$ embryos the dye penetrated the entire ventral aspect (Fig. 8a). At E18.5, the control embryos presented a nearly intact epidermal barrier, whereas in mutants the dye still penetrated in distal limbs (Fig. 8a). The interfollicular epidermis of Ric$^{EKO}$ embryos at E15.5 through E17.5 was consistently thinner compared with controls (Fig. 8b). Hair follicle formation appeared similar in controls and mutants.

To quantify epidermal differentiation markers mRNA and protein was isolated from epidermal tissues at E17.5 and subjected to qRT–PCR and western blotting analysis. Whereas mRNA expression of basal cell markers (K14 and K5) was comparable in mutants and controls, that of differentiation markers K10, K1 and filaggrin was significantly reduced in mutants (Fig. 8c). Expression

of loricrin although reduced in mutants did not reach statistical significant difference compared with controls. Western blot analysis revealed marked reduction of processed Filaggrin in mutants when compared with controls, whereas reduction of K10 was less prominent (Fig. 8d).

To determine downstream targets of mTORC2 in epidermal morphogenesis and to identify factors that may explain the hypoplastic epidermis in Ric$^{EKO}$ mutants, we performed western blotting analysis of the epidermis in E17.5 Ric$^{EKO}$ embryos. Consistent with the attenuation of mTORC2 pathway phosphorylation of Akt at S473 and PKCα at S657 was significantly reduced (Fig. 9a). Phosphorylation of the mTORC2 downstream targets FoxO1-pS256, FoxO1-pT24, GSK3α-p21, and GSK3β-pS9 appeared similar to controls. Also, phosphorylation of the mTORC1 downstream targets S6K-pT389, S6-pS240/244 and 4E-BP1-pT37/46 was comparable to controls (Fig. 9a). Immunohisto-chemistry of E16.5 embryonic epidermis showed that phosphorylation of S6-pS240/244 was easily

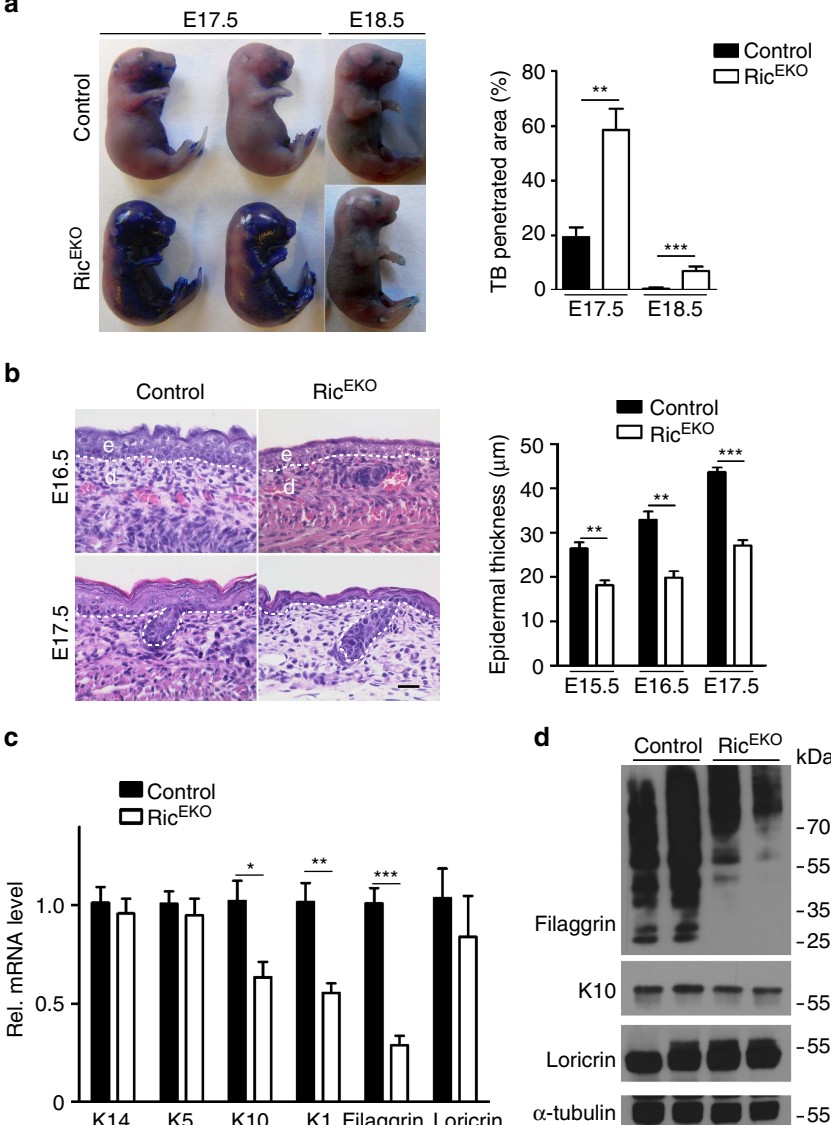

**Figure 8 | mTORC2 regulates epidermal differentiation during embryogenesis.** (**a**) Toluidine blue dye penetration assay of Ric$^{EKO}$ and control embryos at E17.5 and E18.5, and the quantification of the blue stained area ($n = 5$ embryos/genotype). (**b**) Representative H&E-stained back skin of embryos at E16.5 and E17.5 and quantification of epidermal thickness ($n = 5$ embryos/genotype). (**c**) qRT-PCR analysis and (**d**) western blotting analysis of gene expression in Ric$^{EKO}$ and control epidermis at E17.5 ($n = 5$ embryos/genotype). Data represents mean ± s.d.; non-paired $t$-test was used to calculate $P$ value. *$P < 0.05$, **$P < 0.01$, ***$P < 0.001$.

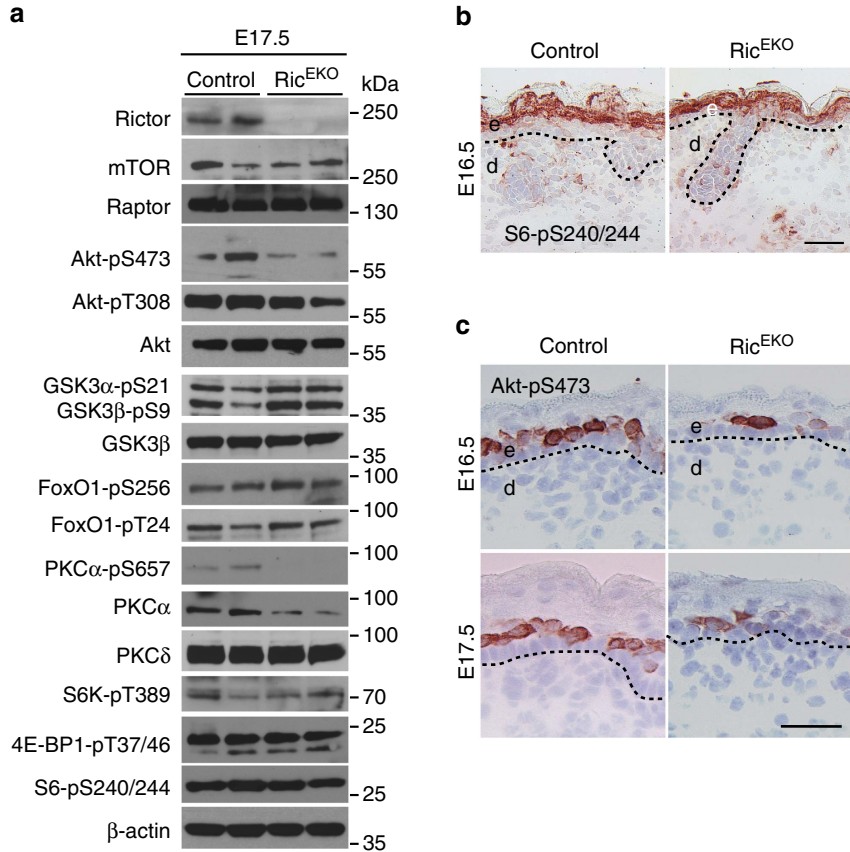

**Figure 9 | mTORC2 signalling is attenuated in Ric$^{EKO}$ mice.** (**a**) Representative western blotting analysis of three independent epidermal extract preparations in Ric$^{EKO}$ mice at E17.5 with antibodies against indicated proteins. (**b,c**) Phospho-S6 (S6-pS240/244) and phospho-Akt (Akt-pS473) immunostaining of back skin in embryos at E16.5 and E17.5. Scale bar, (**b,c**) 25 µm.

detectable in suprabasal cell layers of controls and in Ric$^{EKO}$ embryos, but phosphorylation for Akt-pS473 was hardly detectable in the epidermis of Ric$^{EKO}$ (Fig. 9b,c).

To provide further mechanistic insight into the hypoplastic phenotype in Ric$^{EKO}$ mice we examined several mechanisms that have been reported to contribute to stratification in the developing epidermis[4–6]. First, we performed BrdU pulse labelling studies in mutants and controls during E15.5 and E17.5 to quantify mitotic cells. Whereas at all time points in the basal compartment the number of BrdU$^+$ cells was comparable in control and mutant mice, in the suprabasal layer of E15.5 mutant embryos BrdU$^+$ cells were significantly reduced compared with controls (Fig. 10a). *In vitro* analysis of keratinocytes isolated from newborns did not show alterations in colony forming efficiency or cell proliferation, suggesting that loss of mTORC2 activity in keratinocytes does not cause a cell autonomous proliferation arrest (Supplementary Fig. 5).

During the late stage of development, a switch from symmetric (SCD) towards asymmetric cell division (ACD) in basal keratinocytes plays a central role in initiating the programme of epidermal stratification and differentiation[4–6]. To assess the cellular division pattern we examined the mitotic spindle orientation in basal keratinocytes at E16.5 by staining for the spindle midbody marker Survivin, Par3, a central regulator of polarization processes, the leucine-glycine-asparagine repeat-enriched protein (LGN), and the hemidesmosome marker β4-integrin. Based on the analysis of Survivin staining, quantification of SCD (0–30°) and ACD (60–90°) in controls identified 37% ± 7.6 SCDs and up to 48% ± 6.8 ACDs (Fig. 10b). In contrast, only 24% ± 2.4 of the basal cells in Ric$^{EKO}$ embryos

underwent ACD (Fig. 10b). Furthermore, whereas in basal cells of controls Par3 staining was enriched at the apical side in mutants Par3 staining was disturbed and did not show a distinct localization to the apical side (Fig. 10c). In addition, the number of basal cells in which an apical crescent of LGN had formed was significantly reduced in mutants versus controls (Fig. 10d), indicative of disturbed basal layer keratinocyte polarity. We did not detect major changes in E-cadherin localization or alterations in gene expression of E-cadherin, Gpsm2, Pard3 and Numa1 (Supplementary Fig. 6a,b). Together, these findings suggest disturbed cell polarization and spindle orientation in Ric$^{EKO}$ embryos.

Together, these findings show that the function of mTOR is split between mTORC1 and mTORC2. Whereas mTORC1 mainly controls keratinocyte proliferation within the basal layer, early epidermal stratification, differentiation and epidermal–mesenchymal, mTORC2 primarily controls cell division orientation and late stage barrier formation of the interfollicular epidermis (Fig. 10e).

## Discussion

We have shown that mTOR signalling is essential for proper skin morphogenesis and for the development of a protective epidermal barrier. Loss of epidermal mTOR in mice attenuated keratinocyte proliferation, abrogated the epidermal stratification programme and the formation of hair follicles. Both mTORC1 and mTORC2 contributed to effective epidermal morphogenesis through activation of distinct pathways that functionally cannot compensate for each other (Fig. 10e). Whereas mTORC1 activity

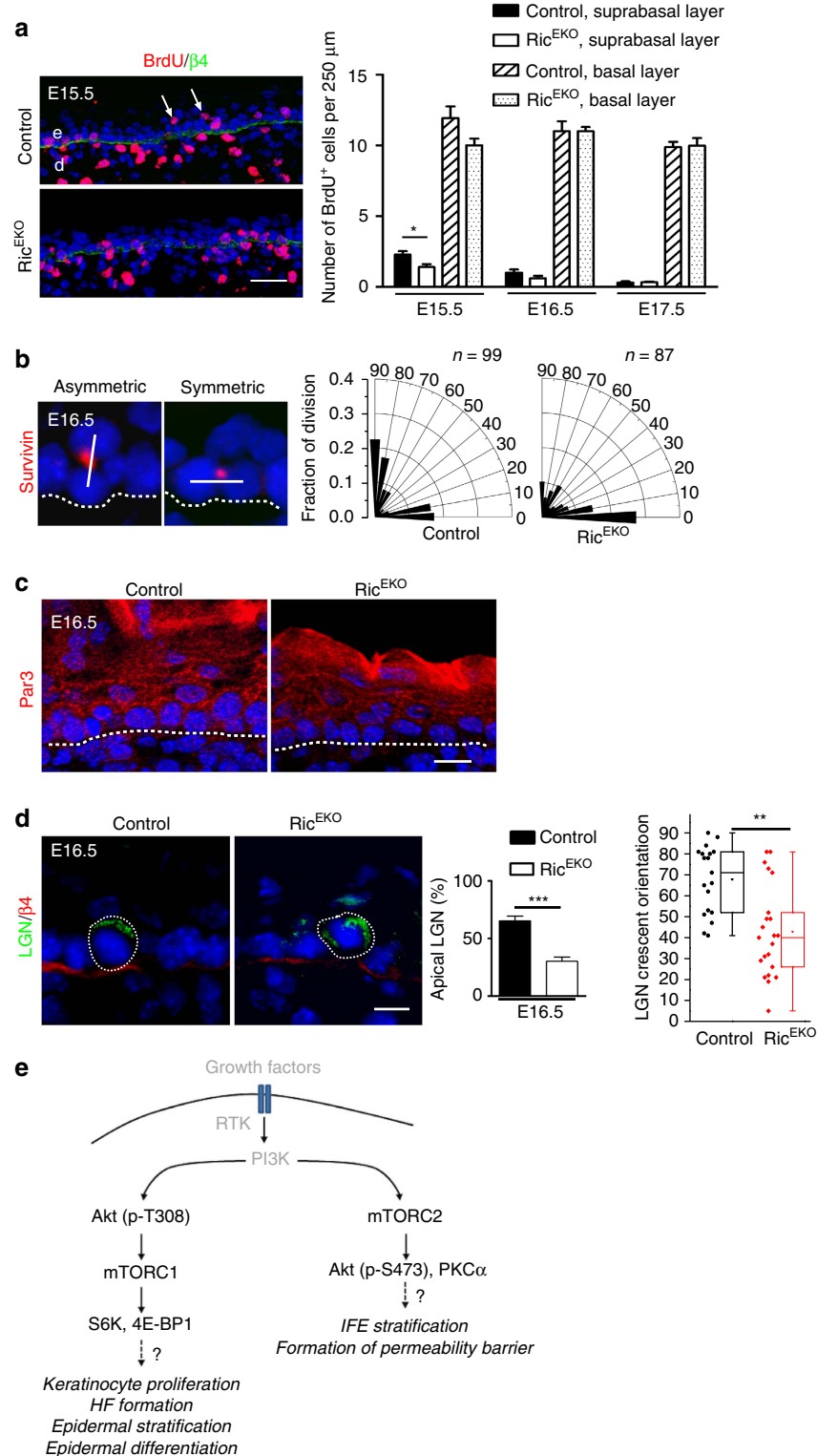

**Figure 10 | mTORC2 regulates asymmetric cell division in embryogenesis.** (**a**) BrdU (red) and β4-integrin (green) double-immunostaing and quantification of basal and suprabasal BrdU$^+$ cells in embryonic back skin ($n = 6$–7 embryos/genotype for one time point). Arrowheads indicate suprabasal BrdU$^+$ cells. (**b**) Survivin-immunostaining (red, DAPI stain blue) to determine basal cell division origination at E16.5 and radial histogram quantification of division angles of basal cells; cell divisions (control: $n = 99$; Ric$^{EKO}$: $n = 87$) were quantified in four embryos per genotype. (**c**) Par3 immunostaining of Ric$^{EKO}$ and control embryonic back skin. (**d**) Immunostaining of LGN in the basal cells of control and Ric$^{EKO}$ epidermis; middle, quantification of basal cells with apical cortical localization of LGN ($n = 5$ embryos/per genotype with $n = 211$ and $n = 159$ cells analysed from control and Ric$^{EKO}$ embryos, respectively); left, dot and box-whisker graph shows LGN crescent orientation in basal cells of E16.5 epidermis. Scale bar, (**a**) 25 μm, (**c**,**d**) 10 μm. (**e**) Hypothetical model illustrating unique functions of mTORC1 and mTORC2 in skin morphogenesis and epidermal stratification. Dashed lines indicate basal membrane. e, epidermis; d, dermis; HF, hair follicle; IFE, interfollicular epidermis. Data represents mean ± s.d.; non-paired *t*-test was used to calculate *P* value. *$P < 0.05$, **$P < 0.01$, ***$P < 0.001$.

is primarily critical during the early phase of epidermal differentiation, mTORC2 exerts its activity at later stages of stratification and terminal differentiation.

The developing epidermis in mTOR[EKO] and Rap[EKO] mice showed remarkable phenotypic similarities, and in both mutants cell proliferation in the basal layer was considerably reduced whereas no signs of apoptosis were detected. The mitogenic function of mTORC1 in the epidermis is consistent with its role in anabolic growth-promoting processes in most cell types, such as stimulation of protein synthesis[11].

S6K and 4E-BP1 are rate-limiting factors in mRNA translation[8,44]. They are the best characterized direct targets of mTORC1 (refs 8,9). We found that phosphorylation of S6K and 4E-BP1 was significantly reduced in the epidermis of mTOR[EKO] and Rap[EKO] embryos, whereas it was not affected in Ric[EKO] embryos. It is likely that reduction of those proteins that control proliferation of epidermal progenitor cells contributed to the hypoplastic epidermis in mTOR[EKO] and Rap[EKO] mutants, and they await identification in future studies. One such protein may be fibroblast growth factor-10 (FGF-10), a mitogenic factor for epidermal homeostasis and in carcinogenesis. It was recently shown that mTORC1-dependent phosphorylation of 4E-BPs is necessary for the activation of translation of FGF-10 mRNA[29]. Furthermore, also perturbed p63 signalling is likely to contribute to the defects in the early differentiation programme observed in the skin of mTORC1 mutants. The delayed switch-off from the simple epithelial keratins K8/18 towards K14 expression, the prolonged maintenance of periderm and attenuated proliferation of basal cells are all indicative for impaired p63 activity in the absence of mTORC1 activity. Our findings of mTORC1 regulated p63 expression is supported by a recent report describing mTOR-induced STAT3/p63/Jagged signalling cascade in cell differentiation[45].

Interestingly, phosphorylation of S6, a downstream target of S6K, was primarily detected in suprabasal layers during epidermal morphogenesis, where cells have withdrawn from proliferation and are committed to terminal differentiation. Our findings are consistent with recent studies in adult mice detecting S6 phosphorylation in suprabasal layers in normal skin, whereas activation in basal layers required a stress response such as wounding or carcinogenesis[27,29]. Together these findings suggest that in epidermal morphogenesis and postnatal epidermal maintenance, S6 may play a primary role in keratinocyte differentiation rather than proliferation. However, a potential function of S6 in epidermal differentiation must be different from our observed mTORC2-mediated effects in keratinocyte differentiation and formation of the cornified envelope because S6 activation cannot rescue epidermal differentiation defects in Ric[EKO] mice. mTORC1-mediated activation of S6-mediated proliferation in keratinocytes might be more important during a disturbed homeostasis such as in wound healing or carcinogenesis. Consistent with this view is also the clinical observation that patients treated with Rapamycin do not develop obvious skin problems unless they are wounded.

The epidermal phenotypes of mTOR[EKO] or Rap[EKO] embryos are strikingly similar to those in IGF-1 receptor (IGF-1R) or Akt1/Akt2 double null mutants. IGF-1-R null mutants display a hypoplastic epidermis and the number of hair follicles is reduced[46]. Similar epidermal defects have also been described in Akt1/Akt2-deficient mice[47]. In a variety of tissues, IGF-1 functions as an extracellular growth signal and plays a pivotal role in activating mTORC1 through the PI3K-Akt signalling pathway[11]. These findings propose a critical role for the IGF-1R-PI3K-Akt1/2 axis upstream of mTOR in epidermal morphogenesis. Notably, the severity of skin defects in mTOR[EKO] and Rap[EKO] mice appeared more pronounced compared with the phenotype in epidermal-specific IGF1R/ Insulin receptor double knockout mice[48], implying the existence of additional signals contributing to mTOR activation in epidermal morphogenesis.

Epidermal-specific deletion of *Rictor* caused a hypoplastic interfollicular epidermis, which although less severely compromised compared with mTORC1-deficient epidermis, was associated with perinatal skin fragility and disturbances in the formation of a functional epidermal permeability barrier, despite the presence of a functional mTORC1 cascade. Together, these findings led us to speculate on a disturbed release of basal cells into the suprabasal layers and subsequent proper epidermal stratification and terminal differentiation. Intriguingly, here we might have identified a previously unrecognized function for mTORC2 in interfollicular epidermal stratification and barrier formation.

The process how during embryogenesis, murine epidermis expands from a single layer of unspecified basal layer progenitors to a stratified, differentiated epidermis is complex and not entirely resolved[2,49]. A multistep mechanism has been proposed. Lechler and colleagues reported on proliferating suprabasal cells during the early phase of stratification until E15.5 that, although rapidly differentiating until the end of gestation, might contribute to the rapid expansion of epidermal layers and the embryo during this final stage of embryogenesis[4]. Furthermore, recently Williams and colleagues unraveled how particularly after E15.5 the interplay between cell polarity and spindle orientation directs epidermal stratification[6]. During the late phase of epidermal development (after E15.5), a shift in the cellular division pattern from SCD towards ACD in basal cells is essential to initiate epidermal stratification, and involves cytoskeleton dynamics and spindle reorientation[2,4–6,48,49].

In Ric[EKO] epidermis we observed multiple cellular alterations that might account for the hypothrophic epidermal phenotype. First, we observed attenuated numbers of BrdU[+] suprabasal cells at E15.5. Secondly, at E16.5 cortical immunolocalization of Par3 and LGN was altered in basal keratinocytes. And third, at E16.5 we detected reduced perpendicular spindle orientation within the stratum basale. How these cellular alterations might converge and contribute to disturbed epidermal stratification and differentiation in Ric[EKO] epidermis remains an important question. Of equal importance is the identification of mTORC2 targets that mediate these stratification programmes and requires a detailed analysis which is however beyond the scope of this manuscript.

Phosphorylation of Akt-S473 is so far one of the best characterized targets of mTORC2 and was clearly induced during late stage epidermal morphogenesis in controls, whereas it was almost absent in the epidermis of Ric[EKO] mice. Our findings suggest that mTORC2 is a critical regulator of Akt-S473 phosporylation, a process that ultimately affects stratification and epidermal barrier function, independent of mTORC1. In liver, mTORC2 has been shown to be critical for Akt-pS473 signalling to FoxO1 and GSK3α/β, which is important for proper hepatic function and maintenance of whole-body metabolism[50]. However, in the developing epidermis of control and Ric[EKO] mice, phosphorylation of FoxO1 and GSK3α/β was comparable, and is unlikely to have contributed to the observed phenotype in Ric[EKO] mice. Along these lines, a recent report revealed a critical role for insulin/IGF-1-dependent control of FoxO1-mediated p63 activation in coordinating ACD and stratification during morphogenesis[42]. The latter finding was corroborated by a significant attenuation of FoxO1-pS256 phosphorylation in IGF-1R-deficient epidermis at E16.5 (ref. 42). Here, we did not detect an alteration of phosphorylated FoxO1 in either Ric[EKO] epidermis or Rictor-deficient cultured primary keratinocytes.

Hence, the mechanisms underlying reduced ACD in Ric[EKO] mice might be independent of FoxO1-pS256.

mTORC2 is also critical for phosphorylation of PKCα-S657 and controlling the stability and activity of PKC proteins[21,51]. Consistently, we showed that the protein level of total and phosphorylated PKCα-S657 was significantly attenuated in Ric[EKO] epidermis and Rictor-deficient cultured keratinocytes. Intriguingly, in a previous study, downregulation of PKCα in an *in vitro* organotypic epidermis model resulted in a hypoplastic epidermis, comparable to the Ric[EKO] phenotype[52]. These findings suggest that mTORC2 controls epidermal stratification through PKCα activation. In line with our hypothesis for a role of mTORC2-PKCα activation in epidermal cell differentiation are recent reports demonstrating that mTORC2-mediated PKCα-pS657 activation regulates tissue differentiation during neuron and mammary morphogenesis[53–55]. These studies revealed that mTORC2 controls actin cytoskeleton rearrangement in neurons and regulates neuron morphology through PKC signalling and the Tiam1-Rac1-PAK-cofilin pathway[53,54]. Actin cytoskeleton rearrangement is important to control epidermal cell shape, shift of cell division orientation and stratification as has been demonstrated in the Srf mouse model[56]. mTORC2 has been shown to play a vital role in the maintenance of the actin cytoskeleton[22,23]. Thus, it is tempting to speculate that mTORC2 promotes ACD and epidermal stratification through regulating the cytoskeleton.

Our findings in skin development corroborate recent reports of a dysregulated mTOR pathway as a pathogenic factor in hyperplastic or inflammatory skin diseases, and hence their therapy with inhibitors of mTOR signalling[57]. However, severe side effects of mTOR inhibition specifically in regenerative responses often require interrupting the therapy, and a more detailed understanding of mTOR signalling in skin biology is critical. Our findings on differential functions of mTOR complexes in skin morphogenesis provide novel insights into their role in skin physiology, and may refine drug development that intervenes in the mTOR pathway.

## Methods

**Animal.** To generate mice with epidermal-specific gene deletion, mice with homozygous floxed *Mtor*[25], *Rptor* or *Rictor*[26] alleles were mated to a *Cre*-transgenic strain expressing Cre recombinase under control of the human K14 promoter[32]. Littermates that either lacked Cre or expressed Cre but carried a heterozygous loxP-flanked *Mtor, Rptor* or *Rictor* allele served as controls. Genotyping was performed by PCR using specific primers on genomic DNA isolated from tail tips as previously described[25,26]. Mice (C57BL/6 background) were maintained and bred under standard pathogen-free conditions. All animal experiments were approved by the national animal care committee and the University of Cologne.

**Quantification of body weight.** Body weight of newborns was monitored every 30 min at room temperature. Weight loss was calculated as percentage of initial weight.

**Skin barrier function assays.** For Toluidine blue staining newborn mice were killed and fixed in methanol for 5 min, washed with PBS and incubated for 5 min in 0.1% Toluidine blue at room temperature; after extensive washing in PBS, images were taken and the blue stained area (indicating non-functional skin barrier) was quantified by using ImageJ software[58]. TEWL was quantified as described previously using a Tewameter[59].

**Histological analysis.** Tissues were fixed in 4% PFA and embedded in paraffin. Sections (10 μm) were stained with hematoxylin and eosin following a standard procedure[60] and analysed using a light microscope (Leica DM4000B, Leica Microsystems, Wetzlar, Germany).

**Immunostaining.** For immunohistochemical and immunofluorescence stainings cryosections from Optimal Cutting Temperature compound (OCT, Tissue Tek) embedded tissues were fixed (4% PFA or in methanol), blocked (10% normal goat serum in PBS) and incubated with primary antibodies (diluted in blocking buffer) over night at 4 °C (ref. 60). Bound primary antibody was detected by incubation with peroxidase-conjugated (EnVision System, Dako) secondary antibody, followed by incubation with peroxidase substrate (Sigma), or Alexa-Fluor 488- or Alexa Fluor 594-conjugated antibodies (Invitrogen). Nuclei were counterstained with hematoxylin or 4′,6-diamidino-2-phenylindole (DAPI, Invitrogen). After washing slides were mounted in mounting medium. Images were taken with a Zeiss Meta 710 Confocal Microscope. Used primary antibodies and their dilutions (Supplementary Table 1): mTOR (Cell Signaling Technology, CST), S6-pS240/244 (CST); Akt-pS473 (CST); K14 (PROGEN Biotechnik); K10 (Covance); K14 (Covance); K15 (Covance); loricrin (Covance); filaggrin (Covance); p63 (Santa Cruz Biotech); P-cadherin (Zymed); active Caspase3 (CST); Survivin (CST); K8/18 (PROGEN Biotechnik); Par3 (EMD Millipore); LGN (EMD Millipore) and β4-integrin (BD Biosciences).

**Electron microscopy.** Freshly isolated skin tissue was fixed in buffer (2% paraformaldehyde, 2% glutaraldehyde, 0.1 M cacodylate buffer at pH 7.35) and postfixed with ruthenium tetroxide[54].

**TUNEL assay.** TUNEL assay was performed on tissue sections using the DeadEnd Fluorometric TUNEL System (Promega) according to the manufacturer's instructions. The nuclei were counterstained with DAPI (Sigma).

**BrdU assay.** Pregnant mice were injected intraperitoneally with 50 μg g$^{-1}$ BrdU (BD Biosciences) 4 h before sacrificing. Tissue samples were OCT embedded and BrdU$^+$ cells were visualized by staining cryosections with a monoclonal anti-BrdU antibody (BD). BrdU$^+$ cells were determined within the stratum basale and suprabasal layers of embryonic back-skin over a distance of 250 μm in five representative fields/section.

**LGN localization and cellular division pattern determination.** LGN apical localization and orientation were determined as described previously[4–6]. Briefly, LGN localization was divided into four quadrants: apical, basal and two lateral quadrants; surface and apical localized LGN was quantified in basal cells of E16.5 epidermis. Measurement of LGN crescent orientation was performed by using the KEYENCE BZ-9000 microscope and BZ-II Analyzer software. The angle of LGN crescent orientation was determined by a line transecting the middle of the LGN crescent through the cell centre relative to the plane of the basement membrane[5].

Cell division orientation was determined as described previously[5]. E16.5 embryo cryosections were stained with surviving to detect anaphase/telophase cells. The cell division angle was assessed by measuring the angle of the plane transecting two daughter cells relative to the plane of the basement membrane. For determining orientation of cell division, radial histograms of cell division angles were plotted in Origin 8.1 (OriginLab). The different divisions were then categorized as described (0 and 30 degrees, symmetric cell division; 30–60 degrees, random; 60–90 degrees, asymmetric cell divisions)[4].

**Keratinocyte culture.** Primary keratinocytes were isolated from newborns and cultured in low calcium medium[60]. For colony-forming assay 3,000 cells were plated in triplicates in collagen-coated 6-well plates (BD BioCoat) and cultured for 14 days in the presence of mitomycin-treated 3T3 feeders[48]. Feeders were changed once a week. Cells were fixed with 4% PFA (20 min) and subsequently stained with 0.1% crystal violet in PBS and photographed. For BrdU incorporation, cells were pulsed with BrdU (10 μM) for 2 or 4 h and BrdU incorporation was analysed by FACS analysis (BD FACSCalibur, BD Biosciences) using the FITC BrdU Flow Kit (BD Biosciences).

**Real-time PCR analysis.** For Real-time PCR epidermis was separated and total RNA was extracted from tissues using RNeasy Minikit (Qiagen), reverse transcription of isolated RNA was performed using the High Capacity cDNA RT Kit (Applied Biosystems)[60]. Amplification reactions were performed with PowerSYBR Green PCR Master Mix (Applied Biosystems) by using 7300 Real Time PCRsystem (Applied Biosystems). The comparative method of relative quantification ($2-^{\Delta\Delta}$Ct) was used to calculate the expression level of the target gene normalized to GAPDH. Primer sequence information can be found in Supplementary Table 2.

**Western blotting.** Epidermis was separated from the dermis and dissociated with a MixerMill homogenizer[42]. For analysis of filaggrin, loricrin and keratin epidermis was lysed in 4% SDS lysis buffer. Alternatively cells or tissues were lysed in radio immunoprecipitation assay (RIPA) buffer, containing protease inhibitor (Sigma-Aldrich) and phosphatase inhibitor (Roche). Protein concentration was determined by Micro BCA Protein Assay Kit (Thermo Scientific) and 20 μg protein per sample was subjected to SDS–PAGE (Invitrogen). Subsequently protein was blotted to PVDF membranes. After blocking (5% non-fat milk in TBST buffer), membranes were incubated with primary antibodies. Primary antibodies and their dilutions included (Supplementary Table 3): mTOR, Raptor, Rictor, Akt, Akt-pS473, Akt-pT308, GSK3α/β, GSK3α/β-pS9/21, FoxO1, FoxO1-pT24/32,

FoxO1-pS265, S6K1-pT389, S6, S6-pS240/244, 4E-BP1-pT37/46 and PKCα (all from Cell Signaling); p63, PKCα-pS657 (Santa Cruz Biotech); loricrin, K10, filaggrin (all from Covance); K14 (PROGEN Biotechnik); β-actin, α-Tubulin (Sigma). After incubation with horseradish peroxidase (HRP)-conjugated anti-mouse, anti-rabbit, anti-guinea pig or anti-goat secondary antibodies (DAKO), the blot was developed with ECL substrate (Pierce) and X-ray film (Amersham Biosciences). Not cropped western blotting results are presented in Supplementary Figs 7 and 8.

**Statistics.** Statistics was performed using PRISM software (Graph Pad Software). Significance of difference was analysed with a Student paired or unpaired two-tailed t-test. All data are presented as mean ± s.d., a P value of ≤ 0.05 was considered significant. The results were presented as the average of at least three independent experiments unless otherwise stated in the legends.

**Data availability.** The authors declare that all data supporting the findings of this study are available within the article and its Supplementary Information Files or from the corresponding author upon reasonable request.

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

## Acknowledgements

The authors thank Michael Piekarek and Sebastian Wüst for excellent technical assistance, and Dr Carien Niessen and colleagues in her group for insightful suggestions and technical advice in TEWL measurements. We thank the CECAD Central Imaging Facility for their support. We thank Dr Sara Kozma and Dr Yann-Gael Gangloff for generously providing the *Mtor*flox mouse line, members of the Eming group and Dr Parisa Kakanj for stimulating discussion. Funding source for this manuscript was the EFRA Program for NRW im Ziel 2, 'Regionale Wettbewerbsfähigkeit und Beschäftigung' 2007–2014 (S.A.E. and L.P.), the CMMC (S.A.E. and M.L.), CECAD (S.A.E.).

## Author contributions

X.D., S.A.E. (conceptualized the study, performed experiments, analysed the data); W.B. (performed TEM analysis); S.I. (helped with the analysis of cell polarity); M.N.H. and M.N.R. (provided *Rptor* and *Rictor* floxed mouse lines); L.P., M.L. (acquisition of funding); all authors (discussed the data and contributed in writing the manuscript).

## Additional information

**Competing financial interests:** The authors declare no competing financial interests.

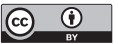

