## [Peer Review File · Nature Communications]

Reviewer #1 (Remarks to the Author)

Ding and colleagues provide detailed descriptions of mouse epidermal phenotypes upon reduced mTOR and associated complex proteins RAPTOR and RICTOR. Reduced function of these proteins produces suites of cellular and structural defects. These changes are well-characterized and show some partitioning between the complex units. The study uses published reagents to reduce protein expression by siRNA in embryos, resulting in early lethality. RNAi is also used in *Drosophila* to knockdown the homologous genes in epidermal tissue. This produces morphological defects in larval cuticle that are broadly described.

While the phenotype analyses of the mouse reduced function genotypes are extensive and professional, the project offers no insights on mechanisms for how these changes are produced by mTOR or either complex cofactor. Nor does the work advance how we understand TOR signaling within cells or in response to extra-cellular inputs. The analysis with *Drosophila* is too unspecific to draw parallels with the mouse phenotypes. It does not present a model for mammalian epidermal biology with respect to TOR. An opportunity for relevance is also lost because the traits studied in either animal are not models of skin pathology. Both cases simply show that interfering with a central regulator of cell growth and maintenance produces defective tissue. This is a limited result. The nice phenotypic descriptions of the mouse mutants may be of interest to specialist in dermal biology.

Reviewer #2 (Remarks to the Author)

This manuscript analyzes the role of mTOR in epidermal keratinocytes. Conditional deletion of mTOR in the epidermis of mice results in postnatal lethality and thin atrophic skin and epidermal appendages such as hair follicles and tongue papillae. Characterization of the epidermis reveals a lack of terminal differentiation and generation of a barrier. Analysis of downstream mTOR targets revealed loss of S6 kinase and 4E-BP1 phosphorylation/activation in mTOR cKO epidermis. The authors also show that loss of Raptor (Rap) but not Rictor (Ric) results in a similar phenotype. Overall, the data presented are convincing and provide a novel picture of the role of mTOR complexes in keratinocytes. These studies significantly extend previous work analyzing the role of Akt in the epidermis. This work will also be of interest to Nature Comm. readers interested in mTOR in other cell types. A few changes to the manuscript outlined below will improve the manuscript prior to publication.

1. The authors should quantify some of the qualitative data presented to give the field a sense of the breadth of the phenotype. In particular, quantification of the barrier phenotype in 3b and 6a, and the epidermal defects in 3c and 6c.
2. The EM images in 3g are hard to delineate the epidermal layers.
3. The authors should also add to either the introduction or discussion the regulation of mTOR signaling by keratin 17 (Kim et al. Nature 2006).

Reviewer #3 (Remarks to the Author)

In this manuscript, Ding and colleagues investigate the roles of mammalian target rapamycin (mTOR) in epidermis development by analyzing the phenotypes of conditional knockout mice with epidermal-specific deletions of mTOR, Raptor (an key component of mTORC1) or Rictor (a key component of mTORC2). This manuscript addresses a very important topic in the field of epidermal development because the roles of mTOR, mTORC1 and mTORC2 have not yet been established in this context.

The authors found that both mTOR and Raptor inactivation during epidermal development are incompatible with the formation of a functional epidermal barrier, with mice dying shortly after birth by rapid dehydration. In contrast, Rictor epidermal ablation causes tissue hypoplasia and transient defects in the epidermal barrier function, which is compatible however with the survival

of mice through adulthood. The authors conclude that mTOR, via mTORC1 plays an essential roles in epidermal morphogenesis, whereas mTORC2 functions are required at early stages of epidermal stratification becoming then dispensable.

Ding and colleagues also use *Drosophila* genetics to explore the conservation during the evolution of mTOR, mTORC1 and mTORC2 functions in epithelial barrier formation. The authors describe similarities between the phenotypes of mice with disruption of mTORC1 and mTORC2, and the phenotypes of analogous *Drosophila* mutants, with severe disruption of epithelial barrier in the case of mTOR or mTORC1 inactivation, and no effects on epithelial barrier following mTORC2 disruption. Although potentially interesting, this latter section of the manuscript does not strengthen the previous section on mouse development because stratification defects represent substantial aspects of the phenotypes of both mTORC1 and mTORC2 conditional mutant mice. *Drosophila* epidermis consists of a simple epithelium. Therefore, and the use of *Drosophila* to investigate developmental functions that in mammalian tissues rely substantially on cell stratification is questionable, and should be presented in a separated study.

The manuscript is overall well written, and the importance of both mTORC1 and mTORC2 in epidermal development emerge clearly from the phenotypic analysis of mutant mice. However, the mechanisms underlying the phenotypes of the different murine models require additional experimental work, since there are several points that are presently unclear in this study.

Specifically:

- 1) The authors describe in mTOREKO epidermis significant defects in the expression levels of mRNA related to keratinocyte stratification and differentiation (Fig S2). It is unclear whether this analysis has been performed using total skins (as stated in Experimental Procedures), or on isolated embryonic epidermal tissue (as suggested in the Fig. S2 legend). If this analysis is performed on total skins at E17.5 without normalization for a housekeeping epithelial-specific gene, this does not provide information on the differentiation state of epidermal cells as it may simply indicate a relative decrease in overall epidermal tissue, which is evident in mTOREKO mice and embryos. Authors should clarify this point. Moreover, because both mTORC1 and mTORC2 are disrupted in these animals, it is essential to establish whether mTORC1, mTORC2, or both, are responsible for this effect by carrying out a similar RT-qPCR analysis in RapEKO- or RicEKO versus Control embryos at E17.5 to clarify which TOR complex is mostly responsible for cell differentiation. This is a key point that needs to be clarified because of the author's claim (First paragraph, page 15 and summary diagram of Fig. 9d) that mTORC1 controls mainly keratinocyte proliferation, stratification and hair follicle formation, while mTORC2 controls cell division orientation, stratification and differentiation. Yet, differentiation protein analysis in RicEKO in P0 epidermis shows no effects of Rictor deletion on K10, loricrin and filaggrin expression levels (Fig. 7g), while the same protein markers appear severely compromised in RapEKO (Fig. 5e), suggesting essential roles of mTORC1 in keratinocyte differentiation control and no contributions of mTORC2, which differs substantially from the authors' conclusions in this matter.
- 2) mTORC1 has key roles in CAP-dependent mRNA translation. The authors should verify whether or not mTORC1 regulates the expression of epidermal differentiation genes in part at the translational level. This should be verified by comparing differentiation marker RNA and protein expression levels in epidermal extracts of RapEKO, mTOREKO and Control newborn mice.
- 3) Regarding the analysis of epidermal differentiation in RicEKO mice at P0, in Fig.7c is visible a significant thinning of the epidermis and in particular of the granular layer, which is correctly pointed out by the authors and quantified in Fig. 7d. However, the positivity for the granular layer markers filaggrin and loricrin appear to be similar to those of Control mice (Fig.7g). Additionally, in several panels of this figure the differences in epidermal thickness between Control and RicEKO mice appear much less pronounced than those shown in Fig. 7c,d. The authors should clarify these inconsistencies.
- 4) In Fig. 8b, the authors show a significant reduction in the thickness of the developing epidermis of RicEKO embryos at both E16.5 and E17.5 relative to Controls. A parallel analysis of both BrdU incorporation and TUNEL staining did not reveal differences between RicEKO and Control embryos

at these stages. However, at E17.5 some BrdU incorporation is visible in suprabasal epidermal cells of Control mice, which is absent instead in the developing RicEKO epidermis (Fig.8C). This may suggest a selective impairment in cell proliferation in the mutant embryos in this compartment, since in the developing epidermis between E15.5-E18.5, epidermal cells can proliferate even suprabasally (see Ref. n.4 of this manuscript). The authors should verify with a more careful analysis whether the reduced epidermal thickness in RicEKO may be coupled with a reduced BrdU uptake in both basal and suprabasal cells. The authors should also better specify how they quantified BrdU incorporation; this is important because they make a strong point in stating that the differences in epidermal thickness between RicEKO and Control embryos and mice are not due to differences in proliferation and/or apoptosis. In Fig. 8d, the authors try to provide mechanistic insight into the hypoplastic phenotype of RicEKO embryos by analyzing the rates symmetric versus asymmetric cell division solely by staining basal epidermal cells for the spindle midbody marker Survivin at E16.5. By itself, this analysis is too simplistic to make the point the authors want to make. The analysis should include membrane markers, cell polarity markers (Par-3-Par6-aPKC) and spindle positioning protein (LGN), and quantification of IF stainings (See Ref. 50 of this manuscript).

5) Moreover, if changes in ACD are the only cause of hypoplasia in RicEKO epidermis (Fig. 8b), what is the fate of basal cells originated by symmetric division in Rictor-deficient animals (Fig. 8d, e) if keratinocytes keep proliferating and surviving normally (Fig. 8C and data not shown). Which events account for the reduced epidermal thickness of these mice?

6) Since the authors do not find changes in mTORC2 downstream signaling molecules that can directly account for the phenotype of RicEKO epidermis and in particular the defects in spindle orientation, the part of the signaling diagram depicted in Fig. 9D concerning mTORC2 is too speculative at this stage of the study.

Response to Reviewers:

Thank you to all three reviewers for your helpful and constructive comments. We greatly appreciate that the reviewers acknowledge scientific interest and novelty of our findings.

We have addressed all of the concerns and believe that the manuscript has been greatly strengthened in doing so. We hope that our work is now acceptable for publication. Changes in the manuscript text are highlighted in yellow.

Reviewer #1:**Point#1:**

“Ding and colleagues provide detailed descriptions of mouse epidermal phenotypes upon reduced mTOR and associated complex proteins RAPTOR and RICTOR. Reduced function of these proteins produces suites of cellular and structural defects. These changes are well-characterized and show some partitioning between the complex units. The study uses published reagents to reduce protein expression by siRNA in embryos, resulting in early lethality. RNAi is also used in Drosophila to knockdown the homologous genes in epidermal tissue. This produces morphological defects in larval cuticle that are broadly described.”

Answer:

We would like to correct the reviewer:

1. We did not use siRNA technology to generate mouse mutants. Epidermal-specific gene deletion in mice was achieved using the Cre-loxP technology (please see page 23, Experimental procedures in the first version of the manuscript).
2. The reviewer is correct in mentioning that we used RNAi to knockdown *dRaptor* and *dRictor* in the larval cuticle in *Drosophila*, however for reducing *dTOR* expression a dominant negative *dTOR* mutant was overexpressed in the cuticle (please see page 15, Results in the first version of the manuscript). This point is now irrelevant for the revised version because as this reviewer and the third reviewer suggested, we decided to present our studies in *Drosophila* in a separate manuscript.

Point#2:

“While the phenotype analyses of the mouse reduced function genotypes are extensive and professional, the project offers no insights on mechanisms for how these changes are produced by mTOR or either complex cofactor. Nor does the work advance how we understand TOR signaling within cells or in response to extra-cellular inputs.”

Answer:

We do not agree with the reviewer. As pointed out in the Introduction, the role of mTOR has not been investigated in epidermal morphogenesis. For the first time our study provides evidence for an essential role of mTOR signaling in epidermal morphogenesis as well as functional dichotomy of mTORCs in epithelial barrier development. The scientific novelty and impact of our findings for the understanding of mTOR signaling in epithelial cells are also clearly acknowledged by reviewer 2 and 3.

Point#3:

“The analysis with Drosophila is too unspecific to draw parallels with the mouse phenotypes. It does not present a model for mammalian epidermal biology with respect to TOR.”

Answer:

We agree with the reviewer that our reported findings in *Drosophila* are quite short; primarily due to space limits of the article. Therefore, we decided to present a more detailed analysis of our findings in flies in a separate manuscript.

However, we do not agree with the reviewer in the point that *Drosophila* cannot be used to

understand biological processes in mammalian epidermis. Many examples in the literature provide evidence that principal mechanisms in epithelial cell function are similar in flies and mammals. Parallel studies in both systems have significantly advanced the field of epithelial biology and were recently nicely reviewed e.g. in an article by Munoz-Soriano et al., *Exp Dermatol* 2014.

Point#4:

“An opportunity for relevance is also lost because the traits studied in either animal are not models of skin pathology.”

Answer:

We do not agree with the reviewer. Numerous examples in the literature document clearly that postnatally restoration of epidermal homeostasis and barrier regeneration after injury recapitulate molecular and cellular mechanisms which are also active during epidermal morphogenesis. Thus, it is expected that our findings in TOR function during epidermal morphogenesis unravel so far unappreciated functions of TORCs in skin physiology and pathology. For example, here we show clearly that phosphorylation of S6K and 4E-BP1 is attenuated in TOR^{EKO} and Rap^{EKO} mutants and might contribute to epidermal stratification defects observed in both mutants lacking TORC1 activity. Activation of both factors has been reported to be altered in skin cancer (Lui *Cancer Discov* 2013; Hertzler-Schaefer *Cell Rep* 2014) or hyperproliferative skin diseases (Ruf *Br J Dermatol* 2014). However, specific contributions of S6K and 4E-BP1 in these skin pathologies is unresolved. Our findings of TORC1 function in epidermal morphogenesis might help to better understand the role of downstream targets in pathology.

Furthermore, in our revised version we provide evidence for reduced filaggrin expression in the granular layer of Ric^{EKO} newborns. These findings open new questions for investigation of TORC2 in epidermal barrier formation and its role e.g. in so far unresolved barrier defects with disturbed filaggrin function e.g. different forms of Ichthyoses.

Point#5:

“Both cases simply show that interfering with a central regulator of cell growth and maintenance produces defective tissue. This is a limited result. The nice phenotypic descriptions of the mouse mutants may be of interest to specialist in dermal biology.”

Answer:

Our manuscript addresses an important topic in the field of epidermal development because the roles of mTOR, mTORC1 and mTORC2 have not yet been established in this context. Again, the scientific novelty and impact of our findings is clearly acknowledged by the other reviewers.

Reviewer #2:

We thank the reviewer acknowledging the scientific novelty of our findings. We are also grateful for the helpful suggestions for improvement.

Point#1:

“The authors should quantify some of the qualitative data presented to give the field a sense of the breadth of the phenotype. In particular, quantification of the barrier phenotype in 3b and 6a, and the epidermal defects in 3c and 6c.”

Answer:

We quantified the barrier phenotypes and epidermal, dermal thickness as suggested. The new data is added to figures 3, 6 and 8.

Point#2:

“The EM images in 3g are hard to delineate the epidermal layers.”

Answer:

We agree with the reviewer and replaced the EM images with new ones in which we delineated the histological structures.

Point#3:

“The authors should also add to either the introduction or discussion the regulation of mTOR signaling by keratin 17 (Kim et al. Nature 2006).”

Answer:

We thank the reviewer for this constructive and important suggestion and included the regulation of mTOR signaling by keratin 17 in the introduction.

Reviewer #3:

We are also thankful to this reviewer for recognizing the importance and novelty of our study, as well as the constructive criticism that we addressed in new experiments.

General comment:

“Ding and colleagues also use Drosophila genetics to explore the conservation during the evolution of mTOR, mTORC1 and mTORC2 functions in epithelial barrier formation. The authors describe similarities between the phenotypes of mice with disruption of mTORC1 and mTORC2, and the phenotypes of analogous Drosophila mutants, with severe disruption of epithelial barrier in the case of mTOR or mTORC1 inactivation, and no effects on epithelial barrier following mTORC2 disruption. Although potentially interesting, this latter section of the manuscript does not strengthen the previous section on mouse development because stratification defects represent substantial aspects of the phenotypes of both mTORC1 and mTORC2 conditional mutant mice. Drosophila epidermis consists of a simple epithelium. Therefore, and the use of Drosophila to investigate developmental functions that in mammalian tissues rely substantially on cell stratification is questionable, and should be presented in a separated study.”

Answer: We followed the suggestion of the reviewer to present our findings in *Drosophila* in a separate manuscript.

Specifically:**Point#1:**

“The authors describe in mTOREKO epidermis significant defects in the expression levels of mRNA related to keratinocyte stratification and differentiation (Fig S2). It is unclear whether this analysis as been performed using total skins (as stated in Experimental Procedures), or on isolated embryonic epidermal tissue (as suggested in the Fig. S2 legend). If this analysis is performed on total skins at E17.5 without normalization for a housekeeping epithelial-specific gene, this does not provide information on the differentiation state of epidermal cells as it may simply indicate a relative decrease in overall epidermal tissue, which is evident in mTOREKO mice and embryos. Authors should clarify this point. Moreover, because both mTORC1 and mTORC2 are disrupted in these animals, it is essential to establish whether mTORC1, mTORC2, or both, are responsible for this effect by carrying out a similar RT-qRT-PCR analysis in RapEKO- or RicEKO versus Control embryos at E17.5 to clarify which TOR complex is mostly responsible for cell differentiation. This is a key point that needs to be clarified because of the author's claim (First paragraph, page 15 and summary diagram of Fig. 9d) that mTORC1 controls mainly keratinocyte proliferation, stratification and hair follicle formation, while mTORC2 controls cell division orientation, stratification and differentiation. Yet, differentiation

protein analysis in RicEKO in P0 epidermis shows no effects of Rictor deletion on K10, loricrin and filaggrin expression levels (Fig. 7g), while the same protein markers appear severely compromised in RapEKO (Fig. 5e), suggesting essential roles of mTORC1 in keratinocyte differentiation control and no contributions of mTORC2, which differs substantially from the authors' conclusions in this matter.”

Answer:

Findings presented in Fig. S2 were generated using total skin as stated in the Experimental procedures. We apologize for the mistake in the Fig. S2 legend.

Therefore, as the reviewer suggested we performed new experiments in which we separated epidermis from dermis at E17.5 from mTOR^{EKO}, Rap^{EKO} or Ric^{EKO} embryos to isolate and analyze mRNA and protein expression primarily in the epidermal tissue.

In epidermis isolated from mTOR^{EKO} (n=5) or Rap^{EKO} (n=5) embryos we normalized gene expression to keratin 14 transcripts. qRT-PCR analysis revealed reduced expression of both basal cell marker K5 and epidermal differentiation markers K1, K10, loricrin and filaggrin. However, whereas expression of K5 was only reduced about 45% and 40% in mTOR^{EKO} and Rap^{EKO} embryos, respectively, expression of the differentiation markers was virtually absent in both mutants (Suppl. Fig. 2a, Fig. 6e). Reduction/absence of differentiation markers was supported by Western blot analysis of epidermal tissues in Rap^{EKO} mutants (Fig. 6f).

In epidermis isolated from Ric^{EKO} embryos (n=5) gene expression was normalized to GAPDH transcripts. qRT-PCR analysis revealed comparable expression of basal cell markers K14 and K5 in controls and mutants, whereas expression of differentiation markers K1, K10 and filaggrin was significantly reduced in mutants compared to controls (Fig. 8c). Interestingly, loricrin expression was similar in controls and mutants. These findings were substantiated by Western blot analysis epidermal tissues (Fig. 8d).

Our new findings corroborate the critical role for TORC1 in epidermal morphogenesis comprising the effective formation of the stratum basale and consequently differentiation of upper cell layers. Further, they highlight that epidermal TORC2 is important for epidermal differentiation as e.g. the effective development of the cornified envelope. One important finding of this study is that TORC1 and TORC2 contribute to effective epidermal morphogenesis through activation of distinct pathways that functionally cannot compensate for each other.

Point#2:

“mTORC1 has key roles in CAP-dependent mRNA translation. The authors should verify whether or not mTORC1 regulates the expression of epidermal differentiation genes in part at the translational level. This should be verified by comparing differentiation marker RNA and protein expression levels in epidermal extracts of RapEKO, mTOREKO and Control newborn mice.”

Answer:

As revealed by H&E histology of mTOR^{EKO} and Rap^{EKO} embryos and newborns, mTORC1 is critical for the development and maintenance of the epidermal basal cell layer. By immunohistochemical analysis only occasionally we observed a keratin 10 positive cell above the stratum basale, whereas proteins of the granular layer (Filaggrin, Loricrin) were entirely absent (Fig. 2d, Fig. 5e). Consistent with the absence of epidermal stratification and effective keratinocyte differentiation, qRT-PCR and Western blot analysis of epidermal tissues of Rap^{EKO} embryos showed a significant reduction of differentiation markers both at the transcriptional and protein level (Fig. 6e,f; see also answer to point #1).

Hence, at this point we do not have any evidence that the disturbed epidermal morphogenesis in mTOR^{EKO} and Rap^{EKO} embryos is specifically caused by impaired mRNA translation.

Point#3:

“Regarding the analysis of epidermal differentiation in RicEKO mice at P0, in Fig.7c is visible a

significant thinning of the epidermis and in particular of the granular layer, which is correctly pointed out by the authors and quantified in Fig. 7d. However, the positivity for the granular layer markers filaggrin and loricrin appear to be similar to those of Control mice (Fig.7g). Additionally, in several panels of this figure the differences in epidermal thickness between Control and RicEKO mice appear much less pronounced than those shown in Fig. 7c,d. The authors should clarify these inconsistencies”.

Answer:

We thank the reviewer for the request to clarify this point. Epidermal hypoplasia in Ric^{EKO} mutants is clearly evident in H&E stainings in newborns and embryos (E17.5) (Fig. 7 c,d). As suggested by the reviewer we complemented the analysis of immunohistochemical stainings for epidermal differentiation markers in Ric^{EKO} mutants by qRT-PCR and protein analysis of isolated epidermis (E17.5) (see also answers to point #1 and #2). These additionally performed analysis showed significantly reduced transcripts for K10, K1, and filaggrin in mutants versus controls (Fig. 8c). Western blot analysis showed a prominent reduction in processed filaggrin when compared to controls (Fig. 8d).

In the immunohistochemical staining for filaggrin, the antibody cannot distinguish between unprocessed and processed filaggrin which is the most likely explanation for the minor difference observed and stated out correctly by the reviewer. Although initially suspected from immunofluorescent stainings, transcripts and protein level of loricrin were comparable in mutants and controls.

Point#4:

“In Fig. 8b, the authors show a significant reduction in the thickness of the developing epidermis of RicEKO embryos at both E16.5 and E17.5 relative to Controls. A parallel analysis of both BrdU incorporation and TUNEL staining did not reveal differences between RicEKO and Control embryos at these stages. However, at E17.5 some BrdU incorporation is visible in suprabasal epidermal cells of Control mice, which is absent instead in the developing RicEKO epidermis (Fig.8C). This may suggest a selective impairment in cell proliferation in the mutant embryos in this compartment, since in the developing epidermis between E15.5-E18.5, epidermal cells can proliferate even suprabasally (see Ref. n.4 of this manuscript). The authors should verify with a more careful analysis whether the reduced epidermal thickness in RicEKO may be coupled with a reduced BrdU uptake in both basal and suprabasal cells. The authors should also better specify how they quantified BrdU incorporation; this is important because they make a strong point in stating that the differences in epidermal thickness between RicEKO and Control embryos and mice are not due to differences in proliferation and/or apoptosis.”

Answer:

We thank the reviewer for this important and constructive suggestion. As the reviewer pointed out Lechler and Fuchs (Nature 2005) reported that at E15.5 epidermal suprabasal layers contain proliferative cells, which potentially contribute to the formation of the multilayered epidermis at this stage. These findings were further substantiated in a recent publication that provided evidence for a two-step mechanism underlying epidermal maturation (Williams et al., Nat Cell Biol 2014).

Following the suggestion of the reviewer we performed additional BrdU pulse-labeling experiments of Ric^{EKO} mutants and controls at E15.5. Interestingly, quantification of BrdU⁺ cells within the basal and suprabasal layers revealed that in E15.5 Ric^{EKO} embryos BrdU⁺ cells were significantly reduced in suprabasal layers when compared to controls. The approach of quantification of BrdU⁺ cells is explained in the revised Material & Methods section.

These more detailed analyses of BrdU⁺ cells suggest that reduced proliferative cells in suprabasal layers during early stratification (E15.5) could contribute to epidermal atrophy in Ric^{EKO} embryos. Quantification of BrdU⁺ cells in suprabasal layers at E16.5 and E17.5 revealed rarely positive cells in controls and in mutant embryos. We added and discussed these findings in the

revised version (Fig. 10).

“In Fig. 8d, the authors try to provide mechanistic insight into the hypoplastic phenotype of RicEKO embryos by analyzing the rates symmetric versus asymmetric cell division solely by staining basal epidermal cells for the spindle midbody marker Survivin at E16.5. By itself, this analysis is too simplistic to make the point the authors want to make. The analysis should include membrane markers, cell polarity markers (Par-3-Par6-aPKC) and spindle positioning protein (LGN), and quantification of IF stainings (See Ref. 50 of this manuscript).”

Answer:

Following the suggestions of the reviewer, we performed new immunohistochemical stainings for E-cadherin, Par3, LGN and β 4-integrin in E16.5 embryos. Confocal imaging analysis revealed that Par3 is enriched at the apical cortex of basal cells in the control embryonic epidermis, but is reduced in basal cells in Ric^{EKO} embryos (Fig. 10c). Furthermore, apical staining for LGN in dividing basal cells, an important element for spindle cell orientation, was reduced in mutants when compared to controls (Fig. 10d). Together, these findings suggest that in Ric^{EKO} embryos polarization and spindle orientation in basal cells is disturbed. Hence, these findings support our observations of reduced ACD in the basal cells of Ric^{EKO} epidermis.

Point#5:

“Moreover, if changes in ACD are the only cause of hypoplasia in RicEKO epidermis (Fig. 8b), what is the fate of basal cells originated by symmetric division in Rictor-deficient animals (Fig. 8d, e) if keratinocytes keep proliferating and surviving normally (Fig. 8C and data not shown). Which events account for the reduced epidermal thickness of these mice?”

Answer:

As outlined in our answer to point#4, based on our findings of reduced BrdU⁺ cells in suprabasal layers in mutant mice, it could be that a combination of both reduced cell proliferation during early stages of epidermal stratification (E15.5) and reduced ACD during later stages contribute to the hypoplastic epidermal phenotype in Ric^{EKO} embryos. We believe this is a highly interesting observation that merits further mechanistic exploration and we also included this point in the discussion. However we would argue that a further detailed experimental investigation of the mechanisms behind TORC2-mediated spindle cell orientation is beyond the scope of this manuscript.

Point#6:

“Since the authors do not find changes in mTORC2 downstream signaling molecules that can directly account for the phenotype of RicEKO epidermis and in particular the defects in spindle orientation, the part of the signaling diagram depicted in Fig. 9D concerning mTORC2 is too speculative at this stage of the study.”

Answer:

We agree with the reviewer that at this stage we cannot yet provide direct and unequivocal functional evidence for the mechanism that accounts for the phenotype in Ric^{EKO} mice. However, in Ric^{EKO} embryos we have clear evidence for a hypoplastic phenotype of the interfollicular epidermis, attenuated suprabasal BrdU⁺ cells in early stage epidermal stratification (E15.5), disturbed spindle cell orientation, and attenuated activation of downstream targets Akt-S473 and PKC α . How these morphological and molecular alterations might be functionally linked remains speculative at this stage. Our findings however provide a solid basis for further mechanistic exploration on how TORC2, so far an entirely unappreciated molecule complex in epidermal stratification and differentiation, might contribute to this complex and so far unresolved process. In light of the current literature we discussed several hypotheses in the revised manuscript (see revised Discussion). To address the reviewer's point we revised the schematic diagram (now Fig. 10e), prudent to avoid any overstatement of our findings.

Reviewer #2 (Remarks to the Author)

The authors have addressed my initial concerns. The additional quantification and figure changes have improved the manuscript. I think the manuscript is now appropriate for publication.

Reviewer #3 (Remarks to the Author)

In this revised version of the manuscript, Ding and colleagues have satisfactorily addressed several of the concerns I have expressed on the previous version of the manuscript. Therefore, in my opinion, the present version has been significantly improved. The main weakness of this study still remains the lack of mechanistic connection between mTORC1 and mTORC2 signaling and the phenotypes observed in the mutant mice. On the other hand, the phenotypic analysis of animals is well performed, and the importance of mTOR signaling in epidermal development emerges from the study, considering that in this context the roles of both mTORC complexes has not yet been determined.

However, I have some important points on the new set of data presented in the revised version that must be further clarified:

1) Regarding the new differentiation marker analysis performed by RT-qPCR on mTOREKO, RapEKO and RicEKO, although in general the results support the concept of a differentiation defect in all mutant embryos, the authors should explain why they normalized data for Keratin 14 transcripts in mTOREKO, RapEKO and for GAPDH in the case of RicEKO. I have suggested the use of the same epithelial housekeeping gene for normalizing data of all three mutant mice versus control mice, to obtain a relative comparison of the magnitude of differentiation defects in epithelial cells. This would also reduce artifacts due to contamination of epidermal tissues with dermal components, which is frequent in early embryonic tissue preparations. GAPDH used for RicEKO mice is far from being an epithelial-specific gene, while K14 expression is likely to be directly affected in both mTOREKO, RapEKO, as indicated by the strong decrease in its protein levels in mTOREKO (suppl. Fig. 2B) and RapEKO (Fig. 6D, F), making its use questionable for normalizing purposes. In fact, if K14 is reduced in the epithelia of these mice, and samples are normalized for its mRNA levels, this will likely generate an overestimate of K5 mRNA levels, and more in general of basal layer markers, the expression of which normally parallels that of K14. K14 and K5 levels are hallmarks of stratified epithelia, and the investigation of the stratification program is central to this study.

One candidate gene for normalizing samples of all mutant mice could be E-cadherin, which is expressed in both simple and stratified epithelia, and whose mRNA levels are similar in RicEKO and Control animals (suppl. Fig 6B).

2) If mTORC1 is essential for establishing early stages of epidermal stratification during embryogenesis, then mTOREKO, and RapEKO epidermal tissues may retain expression of simple epithelia markers such as K8/18. The authors should verify this possibility by RT-qPCR, as it would strengthen the evidence of a developmental arrest and define the fate of cells in which the stratification program aborts in mTOREKO and RapEKO skins.

3) In supplemental Fig. 3 the number of p63-positive cells is strongly diminished in the developing RapEKO epidermis. If this image is representative, then p63 may represent the mechanistic link coupling mTORC1 deficiency with defective epidermal stratification that the authors were looking for. Therefore, they should verify by RT-qPCR the expression of p63 target genes involved in epidermal stratification (e.g. IRF6, etc) in RapEKO epidermis and comment on the results obtained.

Minor points:

In line 282 it should read "were detected in RicEKO pups" .

Response to Reviewer #3:

We thank the Referee for his/her valuable input on the revised data and proposed additional tests to further gain insight into the mechanism of TOR-mediated control of epidermal stratification and differentiation. We believe the additional findings helped to further advance the manuscript.

Point#1:

„Regarding the new differentiation marker analysis performed by RT-qPCR on mTOREKO, RapEKO and RicEKO, although in general the results support the concept of a differentiation defect in all mutant embryos, the authors should explain why they normalized data for Keratin 14 transcripts in mTOREKO, RapEKO and for GAPDH in the case of RicEKO. I have suggested the use of the same epithelial housekeeping gene for normalizing data of all three mutant mice versus control mice, to obtain a relative comparison of the magnitude of differentiation defects in epithelial cells. This would also reduce artifacts due to contamination of epidermal tissues with dermal components, which is frequent in early embryonic tissue preparations. GAPDH used for RicEKO mice is far from being an epithelial-specific gene, while K14 expression is likely to be directly effected in both mTOREKO, RapEKO, as indicated by the strong decrease in its protein levels in mTOREKO (suppl. Fig. 2B) and RapEKO(Fig.6D, F),making its use questionable for normalizing purposes. In fact, If K14 is reduced in the epithelia of these mice, and samples are normalized for its mRNA levels, this will likely generate an overestimate of K5 mRNA levels, and more in general of basal layer markers, the expression of which normally parallels that of K14. K14 and K5 levels are hallmarks of stratified epithelia, and the investigation of the stratification program is central to this study.

One candidate gene for normalizing samples of all mutant mice could be E-cadherin, which is expressed in both simple and stratified epithelia, and whose mRNA levels are similar in RicEKO and Control animals (suppl. Fig 6B).“

Answer:

We agree with the reviewer that using an appropriate epidermal-specific gene as a reference for analyzing gene expression would represent an elegant way of studying the differentiation defects in all three mutants. However, for several reasons to us it appears practically impossible to find such an adequate epithelial marker. The epidermis of mTOR^{EKO} and Rap^{EKO} mutants virtually lacks stratified layers and shows profound developmental defects in the early differentiation program. In contrast, epidermis in Ric^{EKO} mutants demonstrate stratified layers characterized by a late stage differentiation defect. [redacted]

Point#2:

„If mTORC1 is essential for establishing early stages of epidermal stratification during embryogenesis, then mTOREKO, and RapEKO epidermal tissues may retain expression of simple epithelia markers such as K8/18. The authors should verify this possibility by RT-qPCR, as it would strengthen the evidence of a developmental arrest and define the fate of cells in which the stratification program aborts in mTOREKO and RapEKO skins.“

Answer:

In fact, this is an interesting point. Following the suggestion we examined K8 and K18 expression in E17.5 epidermis by qRT-PCR and immunohistochemistry. The mRNA expression of K8 and K18 was normalized to GAPDH. Interestingly, the epidermis from mTOR^{EKO} and Rap^{EKO} skin showed high expression levels of K8 and K18 versus controls. We further used a specific anti-K8/18 antibody and performed co-immunostaining for K8/18 and K14 in E17.5 sections. In control epidermis K8/18 expression was virtually absent. In TOR^{EKO} and Rap^{EKO} mutants K8/18 staining was detected in K14-positive basal cells and suprabasal periderm. The

fact that in the mutants K8/18-K14-double-positive cells are maintained to us suggests that within individual mTORC1-deficient cells the keratin switch is delayed/impaired, and haltet at a stage where both keratins are expressed. Together these findings corroborate that TORC1 inhibition counteracts the switching-off of simple epithelial keratins and results in longer maintenance of periderm.

However, at this stage we believe we have to be cautious to draw any conclusion regarding a potential role of TORC1 in ectodermal - epidermal commitment. We presented (Fig. 3d and Supplementary Fig. 3e) and discussed these new findings in the revised version (pages 8/9, 12, 19).

Point#3:

„In supplemental Fig. 3 the number of p63-positive cells is strongly diminished in the developing RapEKO epidermis. If this image is representative, then p63 may represent the mechanistic link coupling mTORC1 deficiency with defective epidermal stratification that the authors were looking for. Therefore, they should verify by RT-qPCR the expression of p63 target genes involved in epidermal stratification (e.g. IRF6, etc) in RapEKO epidermis and comment on the results obtained.“

Answer:

The severe skin defects in mTOR^{EKO} and Rap^{EKO} mutants are reminiscent of the phenotype in p63^{-/-} mice (Reference 35, 36). To gain more insights into the mechanistic link coupling p63 and the defective epidermal stratification in mTORC1 deficient mice, we performed new experiments and first examined the expression of p63 in E17.5 epidermis by qRT-PCR (Δ Np63) and Western blot. The expression of Δ Np63 isoforms was significantly reduced both mRNA and protein levels when compared to controls. Following the suggestion of the reviewer, we next examined the expression of representative p63 target genes including Irf6, Gata3, Ikk α and Runx2 (Moretti F et al., J Clin Invest 120:1570-7, 2010; Candi E et al., J Cell Sci 119:4617-22, 2006; Koster M et al., Proc Natl Acad Sci USA 104:3255-60 2007). Consistent with p63-activated genes Irf6, Gata3 and Ikk α were down-regulated in mTOR^{EKO} and Rap^{EKO} epidermis compared to controls. In addition, Runx2, one of p63 repressed targets was upregulated in mTOR^{EKO} and Rap^{EKO} epidermis versus controls.

Hence, disturbed p63 expression might indeed contribute to the immature and less stratified epidermis in mTOR^{EKO} and Rap^{EKO} mutants. We show the new data in Supplementary Fig. 2c and Supplementary Fig. 3f and mention them in the results (pages 9, 12).

Minor point:

„In line 282 it should read "were detected in RicEKO pups".

Answer:

We corrected the typo.

[redacted]

Reviewer #3 (Remarks to the Author)

Authors have appropriately addressed the concerns I have expressed on the previous versions of the manuscript, which has been now substantially improved with new experimental data. Therefore, I recommend this manuscript for publication.

Minor point; there is a typo at line 394: "Jagge" should read " Jagged". Please correct.

Response to Reviewer #3:

We thank the Referee for his/her valuable input and for appreciating our additional findings and revision.

„Minor point; there is a typo at line 394: "Jagge" should read "Jagged". Please correct.“

Answer:

We corrected the typo.